# Spatial characterization of long-term hydrological change in the Arkavathy watershed adjacent to Bangalore, India

Gopal Penny[1], Veena Srinivasan[2], Iryna Dronova[3], Sharachchandra Lele[2], and Sally Thompson[1]

[1]Department of Civil and Environmental Engineering, University of California, Berkeley, Berkeley, California, USA
[2]Ashoka Trust for Research in Ecology and the Environment, Royal Enclave Sriramapura, Jakkur Post, Bangalore, Karnataka
[3]Department of Landscape Architecture and Environmental Planning, University of California, Berkeley, Berkeley, California, USA

*Correspondence to:* Gopal Penny (gopal@berkeley.edu)

**Abstract.** The complexity and heterogeneity of human water use over large spatial areas and decadal timescales can impede the understanding of hydrological change, particularly in regions with sparse monitoring of the water cycle. In the Arkavathy watershed in south India, surface water inflows to major reservoirs decreased over a 40 year period during which urbanization, groundwater depletion, modification of the river network, and changes in agricultural practices also occurred. These multiple, interacting drivers combined with limited hydrological monitoring make attribution of the causes of diminishing water resources in the watershed challenging and impede effective policy responses. To mitigate these challenges, we develop a novel, spatially distributed dataset to understand hydrological change by characterizing the residual trends in surface water extent that remain after controlling for precipitation variations and comparing the trends with historical land use maps to assess human drivers of change. Using an automated classification approach with subpixel unmixing, we classified water extent in nearly 1700 man-made lakes, or tanks, in Landsat images from 1973 to 2010. The classification results compared well with a reference dataset of water extent of tanks ($R^2$ = 0.95). We modeled water extent of 42 clusters of tanks in a multiple regression on simple hydrological covariates (including precipitation) and time. Interannual variability in precipitation accounted for 63% of the predicted variability in water extent. However, precipitation did not exhibit statistically significant trends in any part of the watershed. After controlling for precipitation variability, we found statistically significant temporal trends in water extent, both positive and negative, in 13 of the clusters. Based on a water balance argument, we inferred that these trends likely reflect a non-stationary relationship between precipitation and watershed runoff. Independently of precipitation, water extent increased in a region downstream of Bangalore, likely due to increased urban effluents, and declined in the northern portion of the Arkavathy. Comparison of the drying trends with land use indicated that they were most strongly associated with irrigated agriculture, sourced almost exclusively by groundwater. This suggests that groundwater abstraction was a major driver of hydrological change in this watershed. Disaggregating the watershed-scale hydrological response via remote sensing of surface water bodies over multiple decades yielded a spatially resolved characterization of hydrological change in an otherwise poorly monitored watershed. This approach presents an opportunity for understanding hydrological change in heavily managed watersheds where surface water bodies integrate upstream runoff and can be delineated using satellite imagery.

# 1   Introduction

Human water consumption is straining water resources worldwide (Vogel et al., 2015; Gleick, 2014; Wada et al., 2012; Lall et al., 2008), with developing nations particularly vulnerable to water scarcity (Vörösmarty et al., 2010). The causes of water scarcity are complex (Srinivasan et al., 2012) and in south India have been associated with urbanization (Srinivasan et al., 2013), groundwater depletion (Reddy, 2005), degradation of rainwater harvesting structures (Gunnell and Krishnamurthy, 2003), and interstate water disputes (Anand, 2004).

Water scarcity in south India is aggravated by the fact that human activities have shifted or reduced the availability of water resources through inter-basin transfers, artificial conveyance, changes in land use, and irrigation (Mohan and Routray, 2015). Effective management of water resources in south India requires better characterization of the changing nature of water resources (Kumar et al., 2005; Milly et al., 2008) and associated human drivers of change (Venot et al., 2007; Falkenmark et al., 2007; Wagener et al., 2010). Human interventions in the water cycle often occur due to decisions made at local scales, and therefore exhibit considerable spatial heterogeneity when considered at larger scales. This is problematic in this region because most research linking human drivers to hydrological responses focuses on either the local scale (Perrin et al., 2012; Van Meter et al., 2016), or regional to national scales (Gosain et al., 2011; Devineni et al., 2013; Tiwari et al., 2009). There is little research that addresses the emergent effects and heterogeneity of human-driven hydrological change across the watershed scales at which management decisions must typically be made. The gap in scientific understanding at management-relevant scales is strongly associated with lack of data resolution at these scales, and forces water managers to make decisions without sufficient information about cause and effect within watersheds (Batchelor et al., 2003; Glendenning et al., 2012; Lele et al., 2013; Srinivasan et al., 2015).

The data scarcity that challenges understanding of human-driven hydrological change in south India is a common challenge in hydrology and has been extensively explored through the lens of "predictions in ungauged basins" (PUB) over the past two decades (Bonell et al., 2006; Hrachowitz et al., 2013). The methodologies developed through the PUB initiative focused strongly on near-"natural" basins, where proxies for flow behavior (whether climatic, geographic or geomorphic) could be used to form a space in which to extrapolate flows observed in gauged basins to those in the ungauged site (Blöschl, 2013). Extending these techniques to heavily managed catchments presents numerous challenges, including the identification of suitable proxies to define the effects of human intervention and non-stationarity of the water cycle (Thompson et al., 2013). Given the complexity of these managed systems, hydrological reconstruction to infer or reproduce the history of hydrological change can help identify the predominant processes that relate human water use and management with the hydrological response.

Here we present such a hydrological reconstruction covering four decades of extensive hydrological change in the Arkavathy watershed near Bangalore, India (Fig. 1). Concern about water scarcity in the Arkavathy watershed has grown with the loss of historical monsoon-season river flow and reduced inflows to the TG Halli reservoir, which was the primary water supply reservoir for Bangalore between the 1930s and 1970s. These inflows have declined by nearly 80% since the late 1970s, a time period that also included groundwater depletion and loss of storage in surface reservoirs. Analysis by Srinivasan et al. (2015) showed that neither trends in precipitation nor evaporative demand could explain the observed changes in river flow. Instead,

reductions in river channel flow were probably caused by human drivers of change such as expansion of *Eucalyptus* plantations, groundwater depletion associated with irrigated agriculture, and the construction of in-stream check dams (Srinivasan et al., 2015).

Groundwater irrigation grew in popularity in India in the 1960s (Briscoe and Malik, 2006), supplanting tank irrigation in south India in the following decades with the widespread adoption of borewells for groundwater pumping (Janakarajan, 1993a). Groundwater is now the dominant source of irrigation water in the Arkavathy watershed (Lele et al., 2013; Srinivasan et al., 2015). The availability of year-round reliable water supplies led to increases in the extent and intensity of agricultural production, and thus further demand for water. Replacement of traditional crops with *Eucalyptus* plantations, and population growth and urbanization around the periphery of Bangalore, the road network, and other urban hubs have also likely increased water demand. As villages and farmers became more reliant on groundwater, they attempted to augment groundwater recharge by constructing hundreds, if not thousands, of in-stream check dams which impound a portion of streamflow which is then removed from the channel via groundwater recharge or evaporation (Srinivasan et al., 2015). These decentralized land and water management decisions are spatially heterogeneous and characterizing their effects on surface water is hindered by the lack of hydrological records in the Arkavathy. However, spatially explicit characterization of variations in these drivers and hydrological change across the watershed could offer a basis for drawing conclusions about the likely causes of change, thus assisting in the development of management approaches. To date, such analysis has been limited to anecdotal stakeholder accounts (Lele et al., 2013).

Our reconstruction relies on developing a history of change in the end-of-monsoon-season water storage in widely distributed surface rainwater harvesting structures known as tanks (Vaidyanathan et al., 2001; Van Meter et al., 2014). Agriculture in south India was historically sustained by a series of reservoirs known collectively as the "cascading irrigation tank system". Nearly 1700 tanks have been constructed in the Arkavathy watershed. Tanks typically consist of a long, shallow dam bund constructed across a river to harvest surface runoff during the monsoon and supply irrigation water during the dry season. The bund impedes streamflow until the tank fills, overflows, and "cascades" into downstream tanks. Although the dam bunds remain in place, village-level water managers report that the tanks rarely fill up or overflow in large portions of the Arkavathy (ATREE et al., 2015), similar to other watersheds in south India (Janakarajan, 1993b; Gunnell and Krishnamurthy, 2003; Kumar et al., 2016). This decline of tank water is a cause of concern in the Arkavathy and much of the region, and multiple efforts have been initiated to rejuvenate tanks, often without clear understanding of the drivers of degradation of the system (Kumar et al., 2016; Srinivasan et al., 2015).

Other studies have also used small surface reservoirs as aggregators of upstream discharge. For instance, *In situ* measurements of tank water storage have been successfully used to calibrate and validate hydrological models in Andhra Pradesh (Perrin et al., 2012) and Tamil Nadu (Van Meter et al., 2016). Other studies in south India (Mialhe et al., 2008), the USA (Halabisky et al., 2016), Africa (Meigh, 1995; Liebe et al., 2005; Sawunyama et al., 2006; Liebe et al., 2009; Gardelle et al., 2010) and South America (Rodrigues et al., 2012) also use surface water bodies as aggregators of streamflow.

An illustrative example of one of the tanks in the Arkavathy watershed is shown in Fig. 2 for two conditions: one prior to a runoff event, and another following a runoff event in August 2014. This tank, like all tanks in the watershed, is directly

connected to surface flow in the river channel network. Consequently changes in the water surface area within tanks (tank water extent), such as the changes occurring between the two images shown in Fig. 2, provide a proxy for surface flow generation over the upstream catchment area.

Hydrological changes in the Arkavathy watershed should be apparent in historical satellite imagery, as the period of reported hydrological change in the Arkavathy (from the late 1970s onward) coincides with the initial image collection by Landsat satellites in 1972. We develop an automated approach for estimating tank water extent in the Arkavathy watershed using Landsat imagery and apply this approach to reconstruct a timeseries of water extent in tanks from 1973 to 2010. We then undertake a statistical analysis that identifies temporal trends in water extent while controlling for variability in precipitation over the study period. We interpret long-term trends in tank water extent that remain after controlling for precipitation variations as an indication of spatially-variable hydrologic nonstationarity. Specifically, we hypothesize that declines in tank water extent derive from human activities associated with groundwater depletion, such as groundwater abstraction for irrigation or groundwater mining by *Eucalyptus* plantations. To explore this hypothesis, we compare the non-precipitation-related temporal trends of tank water extent against land use profiles developed by Lele and Sowmyashree (2016). These analyses, including remote sensing, modeling of tank water extent, and land use–trend comparison, are outlined in the methods section below.

## 2 Methods

### 2.1 Study site

The Arkavathy watershed spans 4,253 km$^2$ on the western edge of the city of Bangalore in Karnataka, south India (Fig. 1). It has a monsoonal climate and mean annual rainfall of 820 mm. The monsoon season includes the southwest monsoon from June to September and the northeast monsoon from October to December. We therefore refer to April–May as the pre-monsoon period, June–December as the wet or monsoon season, December–January as the end-of-monsoon period, and January–May as the dry season. We also refer to the "monsoon year", analogous to the usual concept of the water year, spanning the period from April to March of the following year. The watershed has a relatively stable daily maximum temperature of 27°C, which peaks near the end of the dry season in April around 34°C, before pre-monsoon rainfall arrives sporadically in April and May. The river is gauged at TG Halli reservoir (Location 2, Fig. 1b) and upstream of Harobele reservoir (Location 5, Fig. 1b).

The watershed contains a mix of urban, natural, and agricultural land uses. Agricultural land can be divided into rainfed grain crops, irrigated vegetable crops, *Eucalyptus* plantations, and other irrigated tree plantations (e.g., areca nut). Most present-day irrigation water in the Arkavathy is sourced from a deep, fractured rock aquifer. Irrigation from tanks is now significant in only a few locations, mostly located downstream of Bangalore. The city of Bangalore imports water from the regional Cauvery river and returns some urban wastewater to the Arkavathy system. Although many tanks are no longer in use, the tank structures remain intact and continue to capture inflow.

The watershed can be divided into 8 subwatersheds (Fig. 3), which include three major tributaries to the Arkavathy (Kumudavathy, Vrishabhavati, and Suvarnamukhi), and 5 other subwatersheds identified by reservoirs or geographic area (Hesaraghatta, TG Halli East, Manchanabele, Kanakapura, Harobele). The major reservoirs in the watershed differ from the tanks in that they

are actively managed, providing water for urban and agricultural water users. For this reason, we focus our analysis of hydrological change on the behavior of tanks.

## 2.2 Remote sensing analysis

The aim of the remote sensing analysis was to generate a timeseries of the surface area of water stored in each tank (referred to from now on as the 'tank water extent') in the Arkavathy watershed. There is minimal rainfall or flow outside the monsoon period, and analysis of tank areas within the monsoon period is inhibited by extensive cloud cover. The analysis therefore focused on end-of-monsoon images from the months of December and January (<5% of rainfall arrives in December).

Landsat satellite imagery was used for analyses, including 16 images taken in December or January between 1973 and 2010 which provided information about end-of-monsoon tank water extent. An additional 32 images were classified to assist in validation, and to provide information about tank water extent variations during the dry season (see Supplemental Material Fig. S1 and Table S1 for imagery dates).

A range of pre-processing and quality assurance and control procedures were performed on the imagery, including converting all Landsat imagery to top-of-atmosphere reflectance (Chander et al., 2009), identifying missing regions of Landsat 1–3 MSS scenes, accounting for the failure of the scan-line corrector (SLC) in Landsat 7 ETM+ images (Scaramuzza et al., 2005; Chen et al., 2011; Catts et al., 1985), and masking of cloud shadows (Zhu and Woodcock, 2012; Irish, 2000; Craven et al., 2002). The location of tanks within the resulting images was determined using a shapefile of tank boundaries obtained from the Karnataka State Remote Sensing Application Centre (KSRSAC, karnataka.gov.in/ksrsac), supplemented by 1970s topographic maps (surveyofindia.gov.in) for the beginning of the study period. The Supplemental Material contains complete information on data sources (Table S2) and pre-processing of imagery (Section S1.1).

The tank water classification method relied on separating pixels containing water from pixels containing land in a spatial region defined by the mapped tank boundaries. Land cover surrounding wetted areas of tanks included vegetation, bare soil, and built-up urban land. We grouped these classes into a single land class, which was characterized by high reflectance in the NIR band and lower reflectance in visible bands (McFeeters, 1996). Water stored in tanks in the Arkavathy watershed varied from clear (with low reflectance in all Landsat reflectance bands) to turbid (more reflective in the visible (Moore, 1980) and NIR bands (Whitlock et al., 1981)). Turbid water exhibited its highest reflectance in the red band due to the red soils in the Arkavathy watershed (Novo et al., 1989). A conceptual representation of the classification algorithm is provided in Fig. 4, and the steps described below are cross referenced to the numbered panels in the figure.

The Normalized Difference Water Index by McFeeters (1996), NDWI = (green - NIR) / (green + NIR), was calculated at a manually-selected reservoir containing clear water (step 1). Otsu's method (Otsu, 1979) was then used to threshold NDWI into land and clear water classes, and the spectral means of both classes were calculated at the training reservoir (step 2). The minimum NDWI of water pixels at the training reservoir (step 3a) was used as a threshold to create a mask of "apparent" clear water for the entire scene (step 3b) which was then dilated using a 5x5 square kernel (a 3x3 kernel for MSS scenes). All pixels within the dilated mask were transformed to a single component, $\hat{x}$, parallel to the transect between the spectral means of clear water and land in the 2-dimensional space of NIR and green reflectance (step 3c). Pixels falling between the $\hat{x}$ means of clear

water and land were assigned a clear water fraction, in the range [0,1], based on the linear distance between the end members along the $\hat{x}$ transect.

A similar procedure of masking, dilating, and unmixing was performed for turbid water, with minor changes. The criteria for apparent turbid water pixels were determined from land pixels near the training reservoir as the 98th percentile of red
reflectance and the 98th percentile of NDWI (step 4a), provided that red reflectance was greater than NIR reflectance. Pixels meeting these criteria were included in the turbid water mask and dilated to include the surrounding area (step 4b). Spectral unmixing was conducted similarly to clear water, except the component for unmixing, $\hat{y}$, was taken along the transect between the spectral means of turbid water and land in the NIR-red space (step 4c). Finally, the water area in each pixel was taken as the higher value of clear water area and turbid water area (step 5). Tank water extent was calculated as the sum of water area
of all pixels within two pixels of the mapped tank boundary (step 6).

We did not estimate the area of water in any tank that was flagged for the following quality concern criteria: (i) spatial overlap or adjacency of dry tank boundary or wetted tank area with clouds or cloud shadows, (ii) spatial overlap of greater than 25% of dry or wet tank area with missing pixels due to the SLC error in Landsat 7 images, or (iii) greater than 25% spatial overlap of dry or wet tank area with the edge of the scene from MSS images (step 7). In each of these cases, the tank area was
recorded as "NA". Examples of the classification and resulting timeseries of tank water extent are shown in the Supplemental Material for a small tank ($\approx$ 25 ha, Fig. S4) and a large tank ($\approx$ 160 ha, Fig. S5).

Remote sensing and spatial processing were scripted in R (R Core Team, 2016) using the raster (Hijmans, 2015), rgeos (Bivand and Rundel, 2016), sp (Pebesma and Bivand, 2005), and rgdal (Bivand et al., 2016) packages, as well as ggplot (Wickham, 2009) for plotting. Watershed delineation and extraction of the cascading tank network were completed in GRASS
GIS (GRASS Development Team, 2016).

## 2.3  Validation of classification method

Classification results were validated against a 5 m resolution LISS IV satellite image from 26 February 2014 using a classified Landsat image from 27 February 2014. The LISS IV image was classified in ENVI software (Harris Geospatial Solutions Inc.) using support vector machine (SVM) classification with four land classes and four water classes. After classification, the
water classes were merged into a single water class and resampled to the resolution of Landsat so that the resulting grayscale classification contained a water fraction in the range [0,1] for each pixel. The classifications were compared at both the pixel scale and tank scale, while ignoring tanks in which there were obvious differences due to the incongruous image capture dates (e.g., cloud cover).

At the pixel level, a traditional confusion matrix is inappropriate for continuous classification data (Congalton and Green,
2009). Thus, we evaluated the error (Landsat water fraction minus reference water fraction) in all pixels within tanks by binning the pixel error into categories representing under-classified (-1 to -0.2), correct (-0.2 to 0.2) and over-classified (0.2 to 1). We further separated pixels into groups by binning the producer (reference) water fraction and user (Landsat) water fraction. We calculated producer's and user's accuracy for each water fraction bin to form both a producer error matrix and user error matrix.

We also used Digital Globe imagery available from Google Earth (Google Earth, 2016) to assess the validity of the classification in normal (680–955 mm) versus wet (>955 mm) precipitation years during the study period. Given the limited availability of these images, we were unable to find a dry-year image (<680 mm) within the study period that was suitable for comparison with a mostly cloud-free Landsat image. We manually delineated 18 tanks in the normal year (2009) and 34 tanks in wet years (2004 and 2005), and compared the manual delineation with classification of Landsat images from the same time period using a linear regression.

## 2.4 Statistical model of tank water extent

We developed a statistical model to identify changes in tank water extent that could be attributed to changes in streamflow production in the Arkavathy watershed. To achieve this, the model should control for drivers of water extent variability other than streamflow. Bathymetric surveys in the Arkavathy watershed indicate that tank water extent is a function of tank volumetric storage (Young et al., 2017). Thus, a volumetric water balance for a tank can be used to consider the drivers of water extent variability, as follows:

$$S(t_2) = S(t_1) + \sum_{t_1}^{t_2}(P - Drainage - ET)A_{tank} + \sum_{t_1}^{t_2}Q_{in} - \sum_{t_1}^{t_2}Q_{out} - \sum_{t_1}^{t_2}Withdrawals, \tag{1}$$

where $S$ indicates tank storage at time $t_2$ when the Landsat image was taken, $S(t_1)$ is the storage in the tank at some prior time $t_1$, $P$ is the precipitation depth over the tank area, $Drainage$ the drainage from the tank floor, $ET$ evaporation from the tank surface area, $A_{tank}$ is the tank surface area, $Q_{in}$ the streamflow entering the tank, $Q_{out}$ the overflows leaving the tank, $Withdrawals$ any anthropogenic withdrawal from the tank itself, and sums are taken from $t_1$ to $t_2$.

In order to use a regression model to infer long-term hydrological change using records of water extent and precipitation data, we make the following assumptions to account for each of the terms on the right side of the water balance:

1. the initial storage $S(t_1)$ can be approximated with zero,

2. variations in $P$, and thus their contribution to variations in $Q_{in}$, can be accounted for by including precipitation as a covariate in the model,

3. variations in $Q_{out}$ can be neglected, for two reasons: first, because watershed managers report that tanks rarely overflow, so $Q_{out}$ can reasonably be approximated as $\approx 0$, and second because any overflow that does occur implies that $S$ is equal to its maximum $S_{max}$, so that variations in overflow cannot contribute to changes in observed $S$,

4. the sum of $Drainage$, $ET$, and $Withdrawal$ fluxes can be treated as a stationary cumulative loss term, and

5. any time trends in tank water extent that remain, having accounted for (1)–(4), indicate the presence of non-stationarity in tank water extents that could not be explained by variability in precipitation.

We confirmed that (1) is reasonable by analyzing carry-over storage across the dry season using 2014 imagery (selected because of high image availability). Carry-over water extent from 2013 monsoon to the start of the 2014 monsoon was $\leq 25\%$ or approximately $\leq 12.5\%$ of end-of-monsoon storage for more than 50% of tank clusters, and $\leq 50\%$ or approximately $\leq 35\%$ of storage for more than 75% of clusters (water extent to volume conversions are based on bathymetric data reported in Young et al., 2017). Tank clusters with the highest carryover storage (as inferred from water extent) were found in urban subwatersheds or hilly sub watersheds in the southern part of the Arkavathy watershed (see Fig. S8). These results suggest that carry-over storage is minimal in most parts of the watershed and that neglecting its effect on tank water extent variability is reasonable.

Variations in $P$ (2) were accounted for using daily rainfall data from 62 gauges from the Directorate of Economics and Statistics, Government of Karnataka (see Fig. S9 for station coverage). Precipitation trends were analyzed using Mann-Kendall non-parameteric tests. Exploratory analysis at the whole-basin scale indicated that tank water extents were most related to precipitation totals from September 1 to the date of Landsat image acquisition. Contemporary observations in the Arkavathy watershed suggest that only the largest or most intense storms generate runoff. The average depth of large storms (>10 mm/day) from September 1 to the date of the Landsat image was used as a metric of extreme rainfall occurrence to account for these observations.

Finally, we accounted for losses by treating the sum of $Drainage$, $ET$ and $Withdrawal$ fluxes as a lumped linear loss term focusing on the end-of-monsoon and early dry season. Previous analysis of monitored locations shows that since the early 1970s, no streamflow occurred in the Arkavathy watershed other than in months when rainfall occurred (Srinivasan et al., 2015), and rainfall was minimal from December 1 onward. Changes in tank water extent from December 1 into the early dry season are therefore dominated by loss terms. We confirmed that these losses were stationary in 6 of the 8 watersheds analyzed by bootstrapping the non-parameteric Mann–Kendall trend tests using classified tank water extents obtained from 27 dry season Landsat images (see Fig. S8).

All analyses proceeded by considering two spatial scales: 8 subwatersheds and 42 smaller hydrologically-connected sub-watershed units, which are referred to as tank "clusters" (Fig. 3). Each cluster contained at least 15 tanks having non-zero water extent in at least 4 end-of-monsoon images (Fig. 3). Aggregated tank water extents for each cluster form the basis for statistical analysis. Aggregating data in this way overcomes some of the challenges associated with a relatively short record and frequently dry tanks, while offering enough spatial resolution to identify variability in trends across the Arkavathy watershed. The analysis excluded reservoirs, because the water extent in a reservoir is also influenced by active management and water transfers. Some tanks were constructed during the study period, and these tanks were excluded from the analysis in any years prior to their construction.

These model features (1)–(5) were incorporated into a multivariate regression with interactions between continuous co-variates and categorical variables (e.g., see Jaccard et al., 1990; Cohen et al., 2003). The covariates used were cumulative monsoon season rainfall (from September 1 onward), denoted $P_{total}$; average depth of large storms during the monsoon season (from September 1 onward), denoted $P_{extreme}$; time delay from the beginning of the end-of-monsoon period (December 1) to the date of Landsat image acquisition, denoted $DSD$ for dry season days; and the year in which the observation was made,

denoted $Year$. The precipitation variables were calculated for each station, interpolated over the entire watershed using the inverse-distance squared approach, and spatially averaged for each cluster.

The $P_{total}$, $P_{extreme}$, and $DSD$ covariates were modeled as fixed effects which interact with the subwatersheds. In other words, the response of the tank water extent to these variables was allowed to vary for each subwatershed, but was assumed to be consistent for the tank clusters within the subwatershed. The year effect was estimated separately for each tank cluster.

The model can be written as follows:

$$A_{cluster,ij} = C_0 + C_{1,k}P_{total,ij} + C_{2,k}P_{extreme,ij} + C_{3,k}DSD_i + B_{1,j}Year_i + e_{ij} \tag{2}$$

The subscripts refer to the Landsat scene ($i$), tank clusters ($j$), and subwatersheds ($k$). Other than the intercept ($C_0$), the fixed effects differ for each subwatershed ($C_{1,k}$, $C_{2,k}$, and $C_{3,k}$) or tank cluster ($B_{1,j}$). The errors for each observation are included as $e_{ij}$.

The model predicts the tank water extent per cluster ($A_{cluster,ij}$), normalized by its maximum. Tank clusters were only analyzed for any given scene if $\leq 30\%$ of the total cluster tank area was missing (due to tanks being omitted for QA/QC purposes in classification, or not having been constructed by the date of analysis). All covariates were centered by subtracting the mean before being input into the model. We confirmed that collinearity between covariates was minimal and did not impact interpretation of confidence intervals or model output using Generalized Variance Inflation Factors (Fox, 2008; Fox and Monette, 1992) (see Supplemental Material Section S2.2 for details). The model performance was assessed using multiple $R^2$ statistics and significance of all effects.

The primary result of interest is the $Year$ effect on tank water extent for each cluster, $B_{1,j}$. This effect represents a temporal trend in total tank water storage over time (as a percent change over time), after controlling for a stationary relationship between tank water storage and the covariates ($P_{total}$, $P_{extreme}$, $DSD$). In the 6 watersheds where dry season losses were stationary, we attribute this change to changing inflows, as all other sources of non-stationarity are controlled for. In the two subwatersheds where a change in the effect of dry season water loss on tank storage was detected, $B_{1,j}$ captures the combined effect of hydrological change and non-stationarity in dry-season tank water losses.

Because the value of $B_{1,j}$ is the key result of interest, additional analyses were performed to confirm its importance. Specifically the model was refit while omitting the $Year$ effect $B_{1,j}$. The performance of the two models (with and without $B_{1,j}$) was compared via $R^2$ metrics. The significance of deviations between the two model predictions was tested using an F-test ($H_0 : B_{1,j} = 0$, $H_A : B_{1,j} \neq 0$, for at least one value of $j$).

## 2.5 Linear regression of streamflow trend against land use

We used four land use maps developed for 1973-74, 1991-92, 2001-02, and 2013-14 (Lele and Sowmyashree, 2016) encompassing the TG Halli watershed, which contains the three subwatersheds upstream of the TG Halli reservoir (TG Halli East, Kumudavathy, and Hesaraghatta) and includes a total of 17 tank clusters. The maps differentiate agricultural land use classes into rainfed crops, irrigated crops, and *Eucalyptus* plantations. Irrigated agriculture in this region is supplied almost exclusively

by groundwater, allowing us to test whether groundwater irrigated crops, increased water utilization by *Eucalyptus* plantations (Srinivasan et al., 2015), both, or neither, are associated with the identified streamflow trend.

In the early 1970s, rainfed agriculture was the primary land use in the TG Halli watershed. Over the study period, many farmers adopted groundwater irrigation and others converted their fields to *Eucalyptus* plantations, which have the potential to mine shallow groundwater or to significantly reduce deep recharge. These land use changes have the potential to reduce surface water flows by depleting subsurface water availability and baseflow over time, likely resulting in a non-stationary streamflow response. This non-stationarity, in conjunction with the relatively sparse availability of land cover data over time, complicated a direct analysis of land use against tank water level. Instead, a space-for-time approach was used to compare the differences in time-averaged land use across each tank cluster to the differences in the $Year$ effect $B_{1,j}$ found for each cluster. We therefore calculate the time-average land use fraction corresponding to irrigated crops ($A_{irrigated,avg}$) and *Eucalyptus* plantations ($A_{Eucs,avg}$) for each of the 17 tank cluster watersheds and regress ($B_{1,j}$) against these these land fractions:

$$B_{1,j} = C_{Eucs} A_{Eucs,j} + C_{irrigated} A_{irrigated,j} \tag{3}$$

The coefficients, $C_{Eucs}$ and $C_{irrigated}$, correspond to the sensitivity of hydrological change to time average *Eucalyptus* land cover and irrigated agriculture land cover, across all 17 tank clusters. This analysis is not designed to directly infer causation, but rather to understand associations between streamflow decline and agricultural practices.

## 3 Results

### 3.1 Accuracy assessment

The Landsat classification performed best for pixels that were fully dry or wet, when compared with the reference (LISS) classification (Fig. 5a). Producer's accuracy was 84% for wet pixels and 99% for dry pixels, and because of the high number of dry pixels the overall accuracy was 98%. Pixels containing a mix of water and land (20–80% water) had lower producer's accuracy (41–82%). Overall, the classification errors were unbiased and the histogram of classification errors (excluding pixels with zero error) was approximately normally distributed (Fig. 5b).

The Landsat classification agreed well with the reference LISS classification at the tank scale, and accuracy improved with increasing tank size. A regression of Landsat extent versus reference extent (Fig. 6) for tanks less than 25 hectares (278 pixels) had a slope of 0.98 and coefficient of determination ($R^2$) of 0.95. When all tanks and reservoirs were included, the regression line had a slope of 1.02 and coefficient of determination of 0.99. Over 99% of dry tanks were correctly classified as dry, but error was considerably large for small tanks with non-zero water extent less than 2.5 ha (28 pixels), due to false positives in the reference classification as well as errors the Landsat classification. For tanks between 2.5 and 10 ha the classification performed considerably better. The mean absolute error increased as the extent of the water body increased, but mean percent error decreased with water body size. Comparison of our automated Landsat classification similarly compared well with the Google Earth manual delineation of tanks in both normal years ($R^2 = 0.97$) and wet years ($R^2 = 0.97$) (see Fig. S6).

Although the time-variation in most tanks have not been reported via *in situ* measurements, trends in water storage over time are widely known for some of the major reservoirs. The TG Halli and Hesaraghatta reservoirs declined from a peak storage in the 1970s to much lower contemporary storage. Large increases in water extent were observed in Manchanabele reservoir, which was constructed in 1993, and Harobele reservoir which was constructed in 2004. These anecdotal trends corroborate our findings for these specific structures (Fig. S7).

## 3.2 Statistical analysis

Trend analysis of the 62 rain gauges in the watershed showed that there were no statistically significant trends in rainfall at whole watershed (see Fig. S10), subwatershed (not shown), or tank cluster scales (see Fig. S11). Precipitation has thus been stationary, although exhibiting considerable inter-annual variability, during the period of analysis, and any identified trends in tank water extent over time can exclude consideration of precipitation change as a driver.

The multivariate analysis explained nearly 70% of the variation in tank cluster water extent ($R^2$ = 0.68). Model residuals were normally distributed (Fig. S12). The effects of both precipitation covariates ($P_{total}$ and $P_{extreme}$) were significant (the 95% confidence interval of the slopes excluded zero) in nearly all subwatersheds, and the effect of dry-season water loss was significant in the two subwatersheds that flow into TG Halli reservoir. Inter-annual variability in precipitation ($P_{total}$ and $P_{extreme}$) explained 63% of the total predicted variability in tank water extent over the study period, while the $DSD$ term explained 10% of the variability. Variability in tank water extent due to precipitation was fairly similar across clusters, while the variability due to temporal trends varied greatly across clusters.

The multivariate analysis identified significant $Year$ effects $B_{1,j}$ (Table S3, Fig. S13) in 13 tank clusters. $B_{1,j}$ varied in its sign and statistical significance among tank clusters, and explained 27% of the total variation in tank water extent. In the two subwatersheds flowing directly into the TG Halli reservoir, $B_{1,j}$ captured the combined effect of non-stationarity in streamflow generation and non-stationarity in dry-season tank water losses (lower tank losses increase $B_{1,j}$). If the sign of $B_{1,j}$ is negative in these tanks, it implies that the effect of non-stationarity in streamflow generation must both be negative and exceed the effects of reduced tank water losses. We converted the units of $B_{1,j}$ to an areal rate of change over time per 10 km$^2$ of catchment area (Fig. 7). In the three subwatersheds upstream of TG Halli reservoir, most tank clusters exhibit negative $B_{i,j}$ values, implying reductions in streamflow generation. Tanks within Bangalore generally exhibited negative $Year$ effects, and tanks at the city periphery and immediately downstream of the city had positive effects. Other regions of the watershed exhibited mixed values of $B_{i,j}$, but none were statistically significant at the 95% confidence level.

We confirmed that the $Year$ effect $B_{1,j}$ was important for understanding the variations in tank water extent. Omitting the $Year$ effect from the tank water extent model lowered the $R^2$ from 0.68 to 0.58. Furthermore, the model predictions with and without the $Year$ effect were significantly different according to the F-test ($p < 3.1 \times 10^{-11}$). These results allow us to reject the null hypothesis that $B_{1,j} = 0$, meaning that the $Year$ effects could not be ignored.

Overall, the results indicate that while (unsurprisingly), interannual variations in rainfall totals and extremes explain the majority of interannual variation in tank water level, that a trend in tank water level is present in several regions of the Arkavathy watershed that is independent of rainfall variability. This trend cannot be explained by trends in rainfall, which were negligible

(Figs. S10 & S11), by trends in dry season tank water loss rates, which, where they existed, had the opposite sign to the identified trend in water level (Fig. S8), or by changes in outflows, which are constrained to occur when tank storage is at its peak. The results suggest that changes in streamflow production independent of rainfall are occurring in discrete locations in the Arkavathy watershed, and that the sign of these changes varies through space.

## 3.3 Streamflow decline and agricultural practices

The regression of the $Year$ effect $B_{1,j}$ on irrigated agriculture and *Eucalyptus* land use areas explained most of the differences in $B_{1,j}$ between tank clusters ($R^2$ = 0.68). The relationship between irrigated crops and $B_{1,j}$ was statistically significant (95% confidence intervals of $C_{irrigated}$ excluded zero), and the relationship with *Eucalyptus* plantations was not statistically significant (Fig. 8).

## 4 Discussion

### 4.1 Long-term hydrological trends and human drivers of change

Tank water extent at the end of the monsoon season can be primarily attributed to the storage of monsoon season streamflow, given that tanks in the Arkavathy watershed rarely overflow, there is little carry-over storage year to year, and loss processes do not extensively deplete the tanks from the end of the monsoon period to the time when tank water extents were observed by Landsat. Thus, storage of water in tanks offers an integrated measure of tank inflows during the previous wet season.

Statistical analysis of the tank water extents suggests that while inter-annual variability in tank water extent is largely explained by precipitation, this variability is superimposed on a longer-term trend in tank water extent that is independent of precipitation, representing a non-stationarity in inflows. Analysis of rain gauges indicated that precipitation has been stationary within the watershed during the study period. Non-stationarity in inflows, coupled with stationarity in precipitation, indicate changes in the runoff ratio (defined as flow production per unit precipitation), a common indicator of changing hydrological processes (Hughes et al., 2012).

Historical land use maps for the TG Halli watershed indicate that there is an association between the inferred streamflow generation trends (particularly streamflow declines) and human drivers of change. We hypothesized that the inferred decline in streamflow would correspond with agricultural practices associated with groundwater depletion. Although little data exist to describe historical declines of the water table, contemporary farmers typically have to drill new borewells to depths exceeding 100 m to reach any groundwater. If a loss of baseflow due to groundwater depletion and the disconnection of the water table from the stream channel is a primary driver of streamflow decline, we would expect the negative trends in streamflow to correspond with irrigated agriculture, which is supplied almost entirely by groundwater in the TG Halli watershed.

In the linear model relating the $Year$ effect $B_{1,j}$ to land use in the TG Halli watershed (Equation 2, Section 1), the time-averaged irrigated crop land use area is a clearer and stronger predictor of declines in tank water extent than *Eucalyptus* land use (Fig. 8). Moreover, other exploratory analyses showed that irrigated crop land use has higher correlation with $B_{1,j}$ ($R^2$ = 0.68,

see Fig. S14) than rainfed crops ($R^2 = 0.5$) and all other land use types ($R^2 < 0.38$). Areas retaining mostly rainfed crops exhibit higher (less-negative) values of $B_{1,j}$, and lower (more-negative) values of $B_{1,j}$ are associated with areas with higher conversion of rainfed crops to irrigated crops. The finding that *Eucalyptus* plantations do not play a major role in streamflow decline is consistent with field experiments, which show that that *Eucalyptus* plantations tend to reduce soil infiltration capacity and therefore would increase infiltration excess runoff (Penny et al., 2015). There could be some relationship between *Eucalyptus* plantations and non-stationary hydrologic processes, but if so it is secondary to that of irrigated crops.

Areas with a high fraction of irrigated agriculture are also likely to contain relatively higher densities of check dams than other land use types, given the desire to recharge diminished groundwater resources. In the absence of datasets describing the spatial distribution and hydrological properties of check dams (or a viable way to develop such a dataset), this analysis is unable to separate the effect of loss of baseflow due to groundwater pumping from the in-stream losses due to check dams. Both processes likely play a role in observed hydrological changes. Recession analyses indicate that the loss of the shallow water table could plausibly explain the observed magnitude of streamflow declines (Srinivasan et al., 2015), and check dams exacerbate the loss of streamflow by converting water in the stream channel to groundwater recharge (Jeremiah et al., 2014).

The most negative values of $B_{1,j}$ and thus the largest inferred reductions in streamflow production occurred in the north-ernmost regions of the Arkavathy where elevation is higher than other areas of the watershed. Although it may appear that the pattern of decline could be related to upstream-downstream processes and the presence or absence of irrigation return flows (Van Meter et al., 2016), we are doubtful that this effect is important in the Arkavathy at present. Indirect evidence (e.g., surveys) indicates that the water table is hundreds of meters below the surface in northern parts of the Arkavathy water-shed (Srinivasan et al., 2015). Furthermore, the relief in the watershed is $\approx 100$ m over a distance of 50 km in the TG Halli watershed, meaning that system-wide return flows connecting upstream to downstream are unlikely.

Urbanization could result in increased streamflow being routed to downstream tanks, due to increases in impervious surfaces, the fallowing of agricultural land in anticipation of urbanization, and reduced consumptive water use. Increased urban water use produces increased urban effluent, which is discharged to the surface channel network where it can contribute to increases in tank water storage downstream. The observed positive $Year$ trends downstream and on the periphery of the Bangalore urban area are consistent with the substantial increases in Bangalore's imports from the Cauvery river, from 185 million liters per day (MLD) in 1974 to 1350 MLD currently (BWSSB, 2017). Additionally, as the city has grown, groundwater pumping for urban areas has increased to an estimated 600 MLD (Lele et al., 2013). About 40% of Bangalore's sewage of 1400 MLD flows to Byramangala reservoir (Jamwal et al., 2015). This has contributed to additional inflows to Byramangala reservoir and more irrigated agriculture directly downstream of the reservoir. Tanks within urban areas can also exhibit drying trends. For instance, tanks may be encroached upon as residential areas expand. Additional urban wastewater inflow can lead to expansion of algae blooms covering the tank water surface, which can appear as a "drying" of the tank in this analysis.

## 4.2 Assessing the classification and model uncertainty

The classification of small tanks in the Arkavathy watershed poses challenges associated with harmonization of different Landsat sensors and the variability in the spectral properties of "wet" tanks due to variations in water quality and vegetation

extent. The classification tends to overestimate the amount of water in dry pixels and underestimate the amount of water in wet and mixed pixels. Because our classification scheme is designed to avoid bias between images taken with different Landsat sensors, we likely sacrifice some precision with sensors from Landsat missions 5–8.

Because these mixed pixels lie at the boundary of the wetted tank area, classification error would be sensitive to geo-registration error in one or both of the Landsat and LISS images. Error could also arise from our specification that water pixels must lie within 60 m of clearly identifiable water bodies, or the assumptions made during spectral unmixing. Although the classification scheme accounted for only two classes, the spectral properties of the land class varied among dry soil, wet soil, sparse vegetation, and irrigated agriculture. Classification of water was complicated by vegetation in tanks, varying degrees of turbidity, and algae blooms in tanks with considerable wastewater inflow.

Errors at the pixel and tank scales are likely unavoidable given the spectral heterogeneity of both land and water pixels. In particular, tanks containing water of variable turbidity, excessive vegetation, or algae blooms are prone to classification errors. Because pixel-scale errors are unbiased, accuracy at the tank scale improves as tank size increases. Error is further mitigated by grouping tanks into clusters in the statistical model.

The uncertainty of the classification ($R^2$ = 0.99 when all water bodies are included) is small compared with the uncertainty of the statistical model ($R^2$ = 0.68). Although the results of our statistical model imply a non-trivial amount of unexplained variation, Gardelle et al. (2010) reported similar performance ($R^2$ = 0.78) for a model relating precipitation and water extent in a single lake, and noted that the correlation was valid only for a nine-year subset of the five-decade study period. The sources of uncertainty include the complex hydrological processes that relate precipitation, streamflow, and tank water storage, as well as the nonlinear and heterogeneous relationship between water extent and water storage, the neglect of pre-monsoon tank water extent in the model, and the non-stationary behavior of dry-season losses in the two northernmost watersheds. Given this uncertainty, results of our analysis are reasonable given the simplicity of the model and the complexity and heterogeneity of the watershed hydrological response.

## 5   Conclusions

The Arkavathy watershed embodies many of the water security challenges confronting southern India. With data limitations hampering the characterization of changing water supplies in the watershed, remote sensing tools provide insights into the history and spatial pattern of change in water availability and hydrological function. We were able to take advantage of a pre-existing "sensing network" provided by the irrigation tank system throughout the Arkavathy watershed. The high number of tanks in this watershed allowed for a comparison of hydrological change with land use at spatial scales appropriate for a first-order analysis.

The analysis reveals that while inter-annual variations in tank water extent are dominated by inter-annual variation in pre-cipitation, an independent time trend in tank water extent occurs for a subset of the watershed. This trend is not spatially homogeneous, but varies in its magnitude and sign among different regions of the watershed. These differences appear to be associated with differing patterns of land use across the watershed. A comparison of the hydrological trends with agricultural

practices within the TG Halli watershed showed that declines in tank water extent over time, controlling for precipitation, are more closely associated with groundwater irrigated agriculture than other kinds of land use, including *Eucalyptus* plantations. This association is consistent with hypothesized effects of groundwater depletion on streamflow generation in the Arkavathy, and with the potential influence of check-dams in fragmenting the surface flow network (Srinivasan et al., 2015). Further in-

vestigation could attempt to attribute the cause of the inferred streamflow decline, either via a more sophisticated statistical analysis considering the many potential drivers of change or via a mechanistic model of catchment hydrological functioning. Ideally such analysis would also separate the relative effects of loss of baseflow due to groundwater pumping and conversion of surface flows to groundwater recharge via check dams.

Surface networks of rainwater harvesting structures are employed in seasonal climates worldwide, whether in cascading tank

systems in southern India and Sri Lanka, or hillslope farm dams in Australia (Callow and Smettem, 2009; Roohi and Webb, 2012), North-East Brazil (Lima Neto et al., 2011; Malveira et al., 2012; de Araújo and Medeiros, 2013; de Toledo et al., 2014), South Africa (Hughes and Mantel, 2010), the US Great Plains (Womack et al., 2012) and China (Xiankun, 2014; Xu et al., 2013). Capitalizing on these networks as proxy indicators of rainfall and streamflow variation, as in the Arkavathy, could prove a valuable approach to circumventing problems of data scarcity and characterizing changing hydrological conditions.

**6  Data sharing**

The results of the Landsat remote sensing classification are available on hydroshare.org (Penny et al., 2017), including geo-referenced tank locations, water extent timeseries for each tank, and the covariates used in the multiple regression of water extent.

*Acknowledgements.*  We thank the ATREE's EcoInformatics Lab for RS/GIS support, including M. Mariappan for help in procuring satellite

imagery. Penny acknowledges support from the NSF Graduate Research Fellowship Program under Grant No. DGE 1106400, the NSF and USAID GROW Fellowship Program. Srinivasan and Lele acknowledges financial support for this research from Grant No. 107086-001 from the International Development Research Centre (IDRC), Canada. Thompson acknowledges NSF CNIC IIA-1427761 for support of ATREE-UC Berkeley collaborations.

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

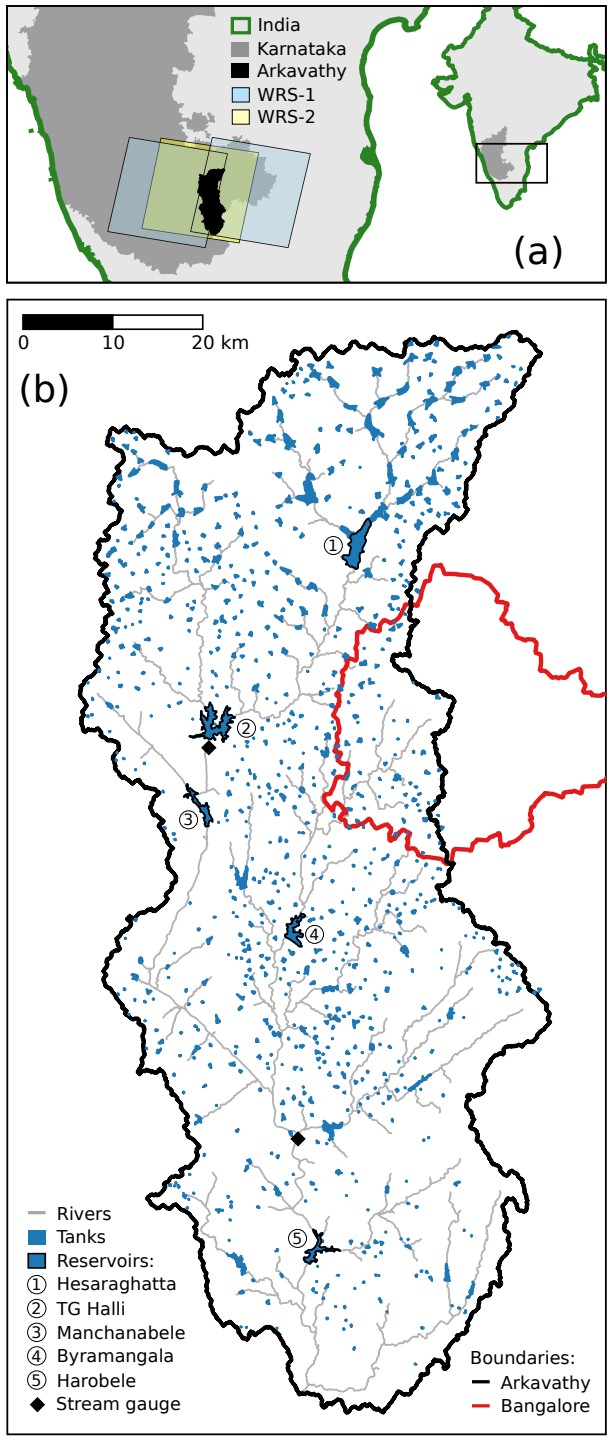

**Figure 1.** Site map. (a) Location of the Arkavathy watershed within the state of Karnataka, India, and scene boundaries for Landsat 1–3 (WRS-1) and Landsat 4–8 (WRS-2). (b) Map of the watershed including tanks, reservoirs including the stream gauge locations, river network, and municipal boundary of Bangalore. Lower-order streams and a number of small, generally dry tanks are excluded.

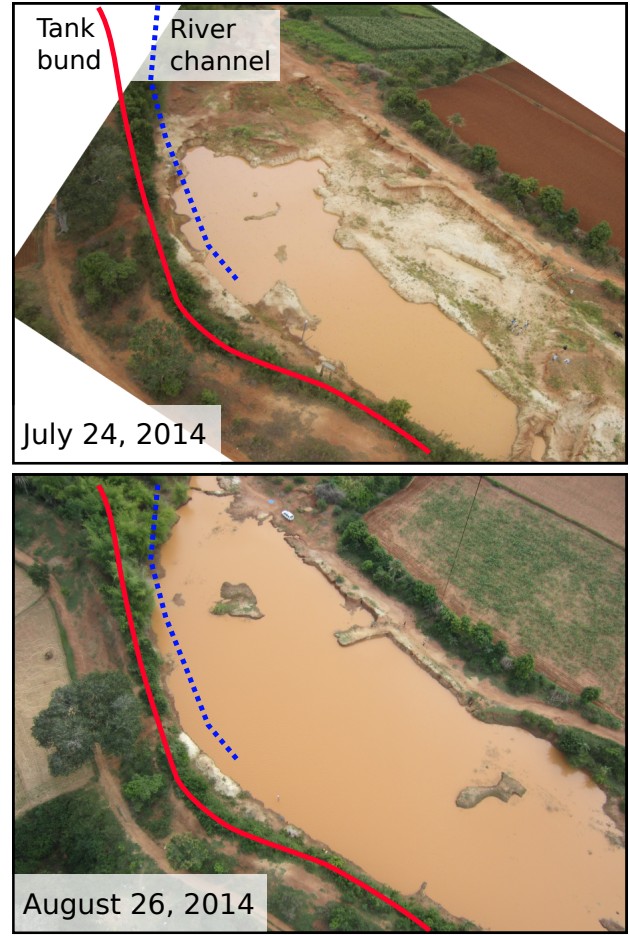

**Figure 2.** Aerial photos of a small tank containing turbid water in the Arkavathy watershed before and after runoff events in August 2014. The tank receives water from the channel and directly from adjacent agricultural plots, and water extent increases with storage.

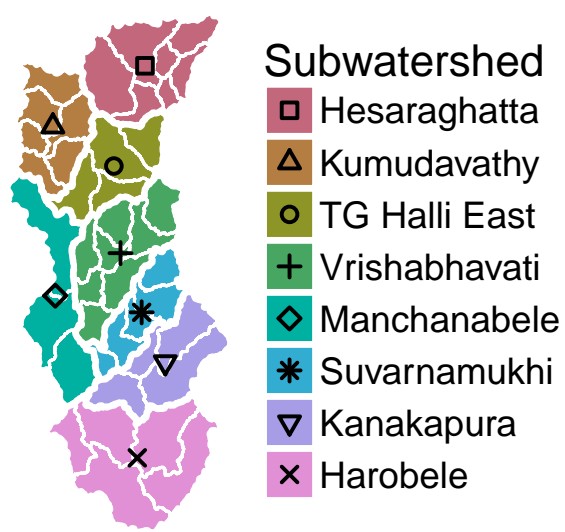

**Figure 3.** Subwatersheds of the Arkvathy watershed. Smaller-scale divisions delineate clusters of tanks (see Section 2.4).

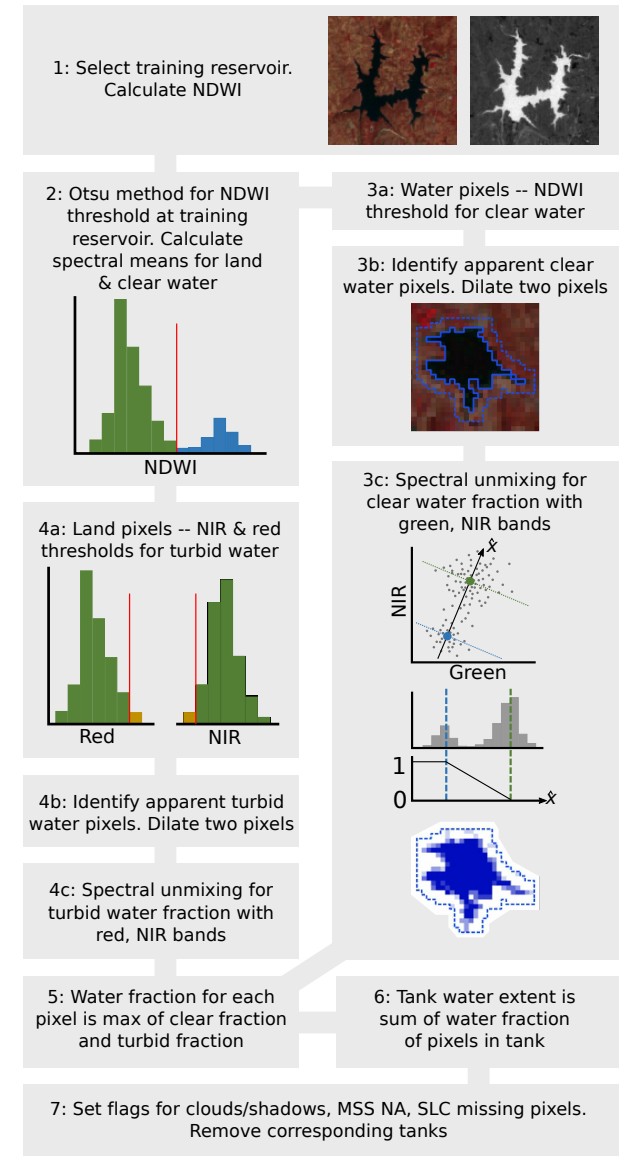

**Figure 4.** Flowchart of classification method. In Steps 3 and 4, clear water fraction and turbid water fraction are each calculated for all pixels in the image before they are combined into water fraction in Step 5. Color images are from Landsat, with red, green, and blue in the image corresponding to NIR, Red, and Green bands from Landsat TM.

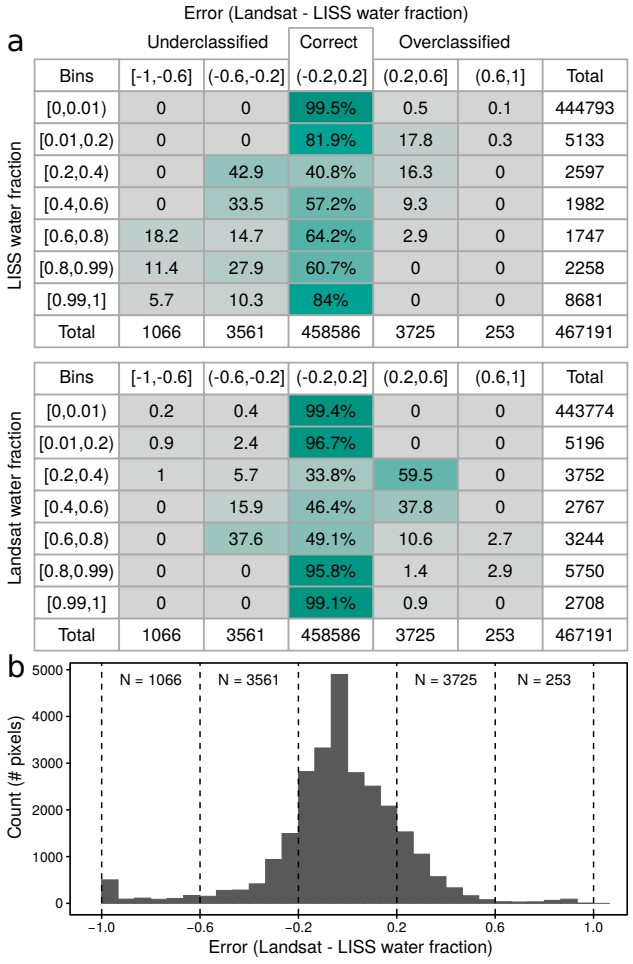

**Figure 5.** (a) Pixel-level producer's and user's accuracy tables, given by percent of pixels within a given error bin. Pixels are grouped into rows by the producer or user water fraction and then binned into columns by the error (Landsat - LISS water fraction). The center column shows the percentage of pixels that were correctly classified, with error between -0.2 and 0.2. (b) Histogram of non-zero classification errors (excluding pixels where the error was zero) with a bin width of 0.0667.

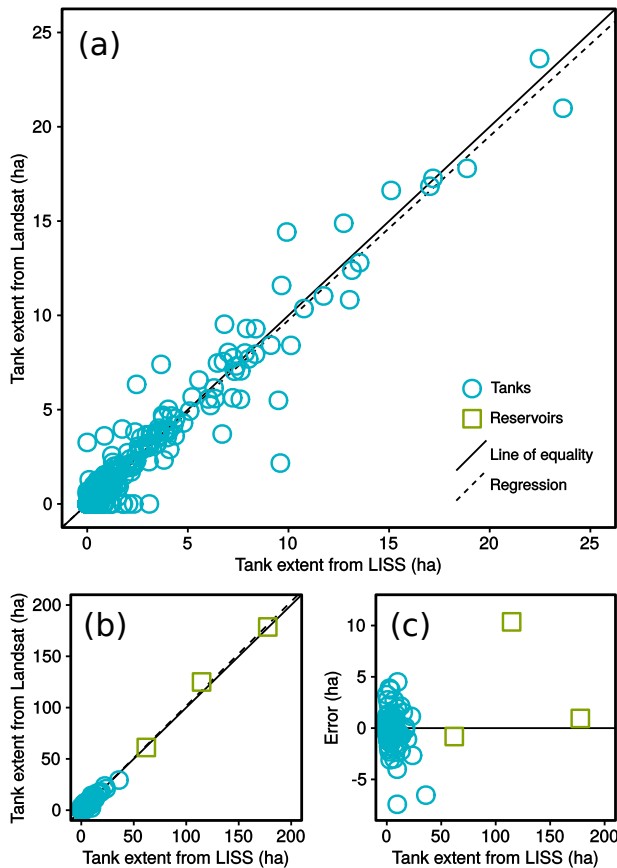

**Figure 6.** Comparison of Landsat and reference (LISS) classification from February 2014 images. (a) Water extent in tanks less than 25 ha. (b) Water extent in all tanks and reservoirs. (c) Error in the Landsat classification for tanks and reservoirs. Relative error decreases with increasing tank size. Only three of the five reservoirs are included because the LISS image excluded the Harobele reservoir and there was considerable change in an algae bloom in the Byramangala reservoir in the time between the acquisition of the LISS and Landsat images.

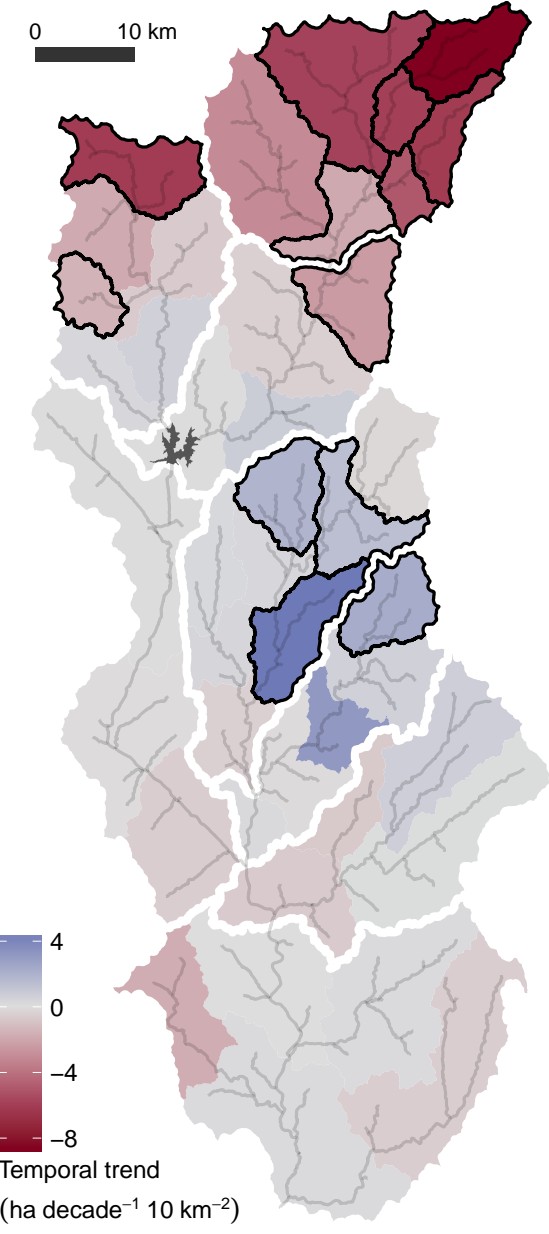

**Figure 7.** Values of $B_{i,j}$, the $Year$ effect on cluster water extent, 1973–2010, given as change in water surface area (ha) per decade per 10 km$^2$ of watershed area. White space indicates subwatershed boundaries, and black lines indicate statistical significance of the cluster trend. Based on analysis of a tank water balance, the sign of $B_{i,j}$ offers insight into likely trends in runoff ratio (streamflow generated within each tank cluster per unit incident rainfall).

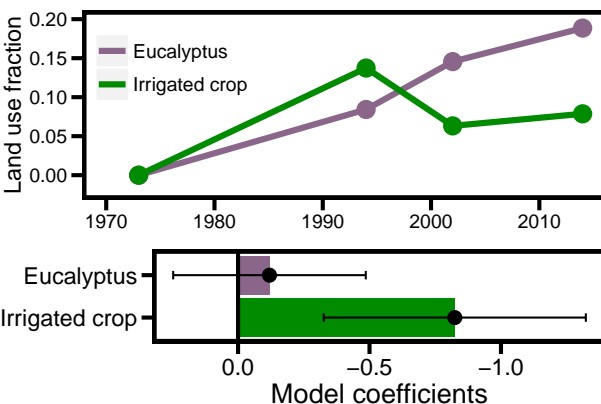

**Figure 8.** Agricultural land use and hydrological change. (Top) Land use fraction of *Eucalyptus* plantations and irrigated crops in four land use maps. (Bottom) Model coefficients ($C_{Eucs}$, $C_{irrigated}$) relating hydrological change to *Eucalyptus* and irrigated crops, based on the multivariate linear regression. Horizontal lines indicate 95% confidence intervals.