# Peer review of "Spatial characterization of long-term hydrological change in the Arkavathy watershed adjacent to Bangalore, India"

_Hydrology and Earth System Sciences, 2016_

## Referee Comment (RC1) · Anonymous Referee #1 · 11 Jan 2017

Strengths

The premise of this paper is interesting and the application of remote sensing to measuring the extent of tank surface area is unique. The paper demonstrates the practical application of remote sensing for characterizing hydrologic change in an otherwise unmonitored setting.

I appreciate the challenge of accounting for various degrees of turbidity in the classification of these water bodies. The methods for measuring tank water extent were clearly presented, and the supplementary figures showing examples of classification were really useful. In my opinion, all of the figures and tables, including those in the supplement, are necessary and contribute to this paper, with the possible exception of

[Figure]

Fig. 6. The supplemental tables should make it possible for someone to reproduce this analysis.

Major Concerns

Although I accept that the multiple regression in Eqn. 1 is a reasonable technique to remove precipitation (climate) effects from the estimate of long-term trend, the analysis of hydrologic change related to land use change is not convincing. The visual comparison of percent agriculture with temporal trend in water extent shown in Fig. 8b does not show a clear relationship. It appears that there is only a temporal trend of magnitude greater than 1 ha decade$^{-1}$ 10 km$^{-2}$ (units should be clarified, is this ha/(decade * 10 km$^2$)?) if the agricultural area is close to 0.75% (which I assume is a typo for 75%); however, low temporal trends are possible for any percent of agricultural land area. This is not a strong argument for a relationship between the two. In fact, the notable negative trends occur only in the two northernmost sub-catchments. An argument could possibly be made that this is an upstream-to-downstream effect, where water withdrawals upstream have a greater impact on stored water over time because return flows from irrigation dampen the effects of water withdrawals in downstream sub-catchments and/or the major reservoirs shown on Fig. 1 are operated in a way that mitigates long-term trends in water storage changes in the tanks (see for example de Graaf et al., 2014). Additionally, as the authors note on p. 7, lines 28-35, the two watersheds farthest upstream (those that drive the trend) were the only two watersheds with a significant trend in dry season water loss, which they relate to the shift from tank irrigation to groundwater irrigation during the study period. Unless I have misunderstood how dry season losses were treated in the regression, this shift would be reflected in the long-term trend. The authors should test whether or not the change in drying rate is the dominant cause of the trend, and if without this shift, a relationship with the % agricultural area still holds. Are there other spatial patterns in rates of groundwater pumping?

The authors develop a simple mathematical model to extract the trend (B) due to "hydrological change", by which I infer that the authors are referring to the "temporal trends in water extent. . .indicative of long-tem hydrological changes induced by human activity" (p. 3, lines 12-13). The intent would be clearer if the authors were to describe other potential causes of this change (for example, temperature change in the region) and to state when defining B in Eqn. (1) that it is the trend (primarily) due to human-induced hydrological change. Also, because dry season loss is a variable in this regression, it is important that the authors clarify exactly which change B is tracking. As described in lines 27-28, p. 7, the dry season loss term is actually the number of dry season days, rather than a volumetric water loss. As such, the trend B presumably includes year-to-year variations in dry season water use as well. This should be stated explicitly, and instead of loss (L) in Eqn. 1, the authors should refer to the variable as what it is, number of dry season days. In summary, the manuscript needs to be more explicit about what exactly the authors intend B to include and exclude, and why.

Secondary Concerns

The one figure that, to me, is basically a throw away is Fig. 6 for multiple reasons. First, the reservoirs are explicitly exclude from all other parts of the analysis, so whether or not their time-trends are correct is immaterial. Second, the figure does not show an independent source of the temporal evolution of reservoir extent. Third, the conclusion that can be drawn from the satellite imagery matching the timing of reservoir construction is simply that the algorithm can distinguish if, in a very large body of water, there is essentially no water or a lot of it. If this were not the case, there would be no merit in even pursuing this approach at all. It would be reasonable to mention that the method shows the timing of reservoir construction and filling as a single sentence.

In terms of reproducibility, it would be helpful if the authors could provide contact information (an address, perhaps) for Karnataka State Remote Sensing Application Centre as a source for a shapefile of tank boundaries in the Acknowledgments section.

p. 7, line 20: please clarify why average depth is used for extreme precipitation events

rather than total number of extreme events or total depth of precipitation in extreme events.

p. 7, last paragraph: reference Fig. S8.

p. 8, 2nd paragraph: define variable terms explicitly (i.e., The covariates total precipitation, Ptotal,ij, . . .) here, close to the equation, instead of in previous paragraphs. State near the equation that the loss is actually the number of dry season days

p. 8 line 19: clarify what is meant by "centered" (long-term means removed?).

Fig. 7: it would be useful to overlay a drainage/stream map to show how sub-watersheds relate.

p. 10, line 1: clarify what is meant by "The spatial scales of tank clusters are comparable with that of land use"

p. 10, lines 16-17: quotes around "drying" make sense because this is referencing algae blooms giving the false appearance of smaller tank water extent. Quotes around "wetting" do not make sense because the increase in impervious surfaces actually causes tank water extent to increase. It may not be more water in the watershed, but it is more water in the tanks.

p. 10, line 29: instead of saying ". . .by focusing on land use from a single date.", say ". . .because we only consider land use on [Mon. Day, Year]"

Figs. S4-S5: at least mention in the caption the water extent vs. precipitation plots.

References

de Graaf, I.E.M., L. P. H. van Beek, Y. Wada, M. F. P. Bierkens, 2014: Dynamic attribution of global water demand to surface water and groundwater resources: Effects of abstractions and return flows on river discharges, Adv. Water Resour., 64, 21-33, doi:10.1016/j.advwatres.2013.12.002.

---

## Referee Comment (RC2) · Anonymous Referee #2 · 12 Jan 2017

Overall this is a well written manuscript that attempted to describe trends and spatial differences in changes in hydrology in the Arkavathy watershed on the basis of changes in extracted tank water surface area from satellite images along with other attributes.

Although the methods were well described, the broader perspective of the analysis is not well presented. After all the study analyzed the tank's surface water dynamics for a very small area (the total area of the Arkavathy is not provided), so, what new information does the findings bring to the community compared to the known facts at regional to national scale for India?

Given the size of the tanks studied, I would imagine the seasonal water area dynamics will have greater implications than the inter-annual dynamics. The manuscript did not

discuss anything on the seasonality for these tanks, or how does that influence the trend?

The manuscript mentioned about differences in water quality, turbidity, vegetation in the water which are influential factors for changes in the reflectance. Even though the DN values were converted to reflectance, the manuscript used only one index (NDWI) to classify water surface area, while there were potentially many other methods or index (Senay et al., 2013) could be used to map water surface correctly, as no one index can cover it all.

While the analysis was performed for the time period between 1972 and 2010 the validation was done for 2014 results. To me validation needs to be done for the time for which the trend analysis is performed (few sample years both wet and dry between 1972 and 2010).

As the study area is so small Google earth might provide good data for validation. Have the authors looked into google earth images as a potential source of validation data?

Page 10 line 5: claims that MK analysis confirms an increase in agricultural land use fraction is related to decrease in tank water storage. How? There is no evidence shown in the manuscript that suggests agricultural land use is increasing. This is vague to me.

Page 10 line 11-12: statement connects with changes in land use and management practice with depleted subsurface stores without providing evidence.

Page 11 line 6-7: Target for classification is to identify water and not water cells, in that case how does incorporation of additional land cover will reduce the classification error?

I think the method used in the manuscript is too simplistic, although producing time-series information of tank water surface area is valuable. I am not sure how much new information has been brought to the community by this study; therefore I am not convinced that HESS is the right journal for this article.

Senay, G.B., Velpuri, N.M., Henok, A., Pervez, M.S., Asante, K.O., Gatarwa, K., Asefa, T., & Jay, A. (2013). Establishing an operational waterhole monitoring system using satellite data and hydrologic modelling: Application in the pastoral regions of East Africa. Pastoralism: Research, Policy and Practice, 3, 20.

———————————————

---

## Author Comment (AC1) · 4 Feb 2017

Thank you for your careful consideration and analysis of the manuscript. Your comments included a number of valid points that we believe will strengthen the paper. Please see the comment titled "Author comments to the editor and referee responses" for our response to your comments.

Please also note the supplement to this comment: http://www.hydrol-earth-syst-sci-discuss.net/hess-2016-562/hess-2016-562-AC1-supplement.pdf

[Figure]

**Supplement:**

*Spatial characterization of long-term hydrological change in the Arkavathy watershed adjacent to Bangalore, India -- Response to Referee 1*

**Response to Referee 1**
Referee comments in black
Our (author) response in blue

Strengths

The premise of this paper is interesting and the application of remote sensing to measuring the extent of tank surface area is unique. The paper demonstrates the practical application of remote sensing for characterizing hydrologic change in an otherwise unmonitored setting.

I appreciate the challenge of accounting for various degrees of turbidity in the classification of these water bodies. The methods for measuring tank water extent were clearly presented, and the supplementary figures showing examples of classification were really useful. In my opinion, all of the figures and tables, including those in the supplement, are necessary and contribute to this paper, with the possible exception of Fig. 6. The supplemental tables should make it possible for someone to reproduce this
Analysis.

- We thank the referee for offering careful consideration and analysis of the manuscript. The referee brings up a number of valid points that we believe will strengthen the paper. We intend to incorporate a number of the suggestions from this referee. We will move Figure 6 to the supplementary material. We will move figures S4 and S5 from the supplementary material to the Results section of the manuscript (excluding the comparisons of water extent and precipitation, which will remain in the supplementary material).

Major Concerns

Although I accept that the multiple regression in Eqn. 1 is a reasonable technique to remove precipitation (climate) effects from the estimate of long-term trend, the analysis of hydrologic change related to land use change is not convincing. The visual comparison of percent agriculture with temporal trend in water extent shown in Fig. 8b does not show a clear relationship. It appears that there is only a temporal trend of magnitude greater than 1 ha decadeˆ(-1) 10 kmˆ(-2) (units should be clarified, is this ha/(decade * 10 kmˆ2)?) if the agricultural area is close to 0.75% (which I assume is a typo for 75%); however, low temporal trends are possible for any percent of agricultural land area. This is not a strong argument for a relationship between the two. In fact, the notable negative trends occur only in the two northernmost sub-catchments.

- Thank you for this comment. Our intention in presenting Figure 8b was to conclude the manuscript with some initial ideas about the attribution of hydrologic change to potential drivers. We agree, however, that despite the statistically significant Mann-Kendall trend, the proportion of a watershed covered with agricultural land use has a tenuous

relationship with hydrological change (at a minimum, one might argue that using a snapshot of land use to explain a decadal trend in hydrology poses a problematic mismatch). We agree that removing Figure 8b is appropriate. We will replace it with a written discussion that offers a broader context for the observed changes. Furthermore, using recently developed maps of land use from 1973, 1994, 2001, and 2013 of the northern Arkavathy watershed (the 3 northernmost subcatchments), we will explore a more detailed analysis of the relationship between land use and hydrological change and present any additional findings in the updated manuscript.

An argument could possibly be made that this is an upstream-to-downstream effect, where water withdrawals upstream have a greater impact on stored water over time because return flows from irrigation dampen the effects of water withdrawals in downstream sub-catchments and/or the major reservoirs shown on Fig. 1 are operated in a way that mitigates long-term trends in water storage changes in the tanks (see for example de Graaf et al., 2014).

- Thank you for this interesting suggestion. Our current hypothesis is that the drying of the northern part of the watershed is linked to groundwater pumping that caused a disconnection between groundwater and surface water (see Srinivasan et al., 2015), leading to reduced baseflow, in a manner analogous to the model in de Graaf et al. (2014). Field studies addressing this mechanism are also currently in progress. Unlike the model presented by de Graaf, however, we are doubtful that the upstream-to-downstream effect is important in the Arkavathy today. Various sources of indirect evidence indicate that the water table is hundreds of meters below the surface in northern parts of the Arkavathy watershed (Srinivasan et al., 2015), suggesting that excess infiltration water is likely to move vertically. Similarly, the relief in the watershed is only about 100 m over a distance of 100 km, again promoting vertical groundwater movement and system-wide return flows connecting upstream to downstream are unlikely. We will make a note to this effect in the revised manuscript.

Additionally, as the authors note on p. 7, lines 28-35, the two watersheds farthest upstream (those that drive the trend) were the only two watersheds with a significant trend in dry season water loss, which they relate to the shift from tank irrigation to groundwater irrigation during the study period. Unless I have misunderstood how dry season losses were treated in the regression, this shift would be reflected in the long-term trend. The authors should test whether or not the change in drying rate is the dominant cause of the trend, and if without this shift, a relationship with the % agricultural area still holds.

- Thank you for this salient observation. Non-stationarity in the dry-season water loss term would indeed affect the magnitude of estimated hydrological change in tank clusters given that the regression relationship used to identify this change assumes a stationary loss coefficient.
- The violation of this stationarity assumption in 2 tank clusters might be expected to marginally increase the model error, and, if the time trend in the dry season losses was

aligned with the time trend in tank storage, it could indeed confound interpretation of the meaning of the storage trend. However, the trend in dry season losses in the northernmost tank clusters is, instead, in the opposite direction to the trend in storage. Dry-season loss rates have *decreased* over time in the two northernmost subwatersheds. We would expect this change to result in an *increase* in tank water storage after monsoon season (as tanks lose water more slowly). Yet we observe a statistically significant decrease in post-monsoon tank storage over time, in spite of the decrease in loss rates. Thus, introducing a non-stationary loss coefficient into the model might improve model fit (at the expense of the degrees of freedom of the model) and improve quantitative estimates of the rate of drying due to hydrologic change in the northern watersheds, but would not alter the main conclusion of the study, which is that these watersheds are, in fact, drying.
- We will add clarification of these points to the discussion.

Are there other spatial patterns in rates of groundwater pumping?
- Understanding the spatial patterns of groundwater extraction in the Arkavathy Basin would be very useful. Unfortunately, monitoring of groundwater use through space has been indirect and sparse. We are exploring whether proxies for groundwater irrigation could be developed from the remote sensing record as an ongoing project, but at this point we are not in a strong position to analyze the effects of such spatial variation on the spatial differences in surface water trends.

The authors develop a simple mathematical model to extract the trend (B) due to "hydrological change", by which I infer that the authors are referring to the "temporal trends in water extent: : :indicative of long-tem hydrological changes induced by human activity" (p. 3, lines 12-13). The intent would be clearer if the authors were to describe other potential causes of this change (for example, temperature change in the region) and to state when defining B in Eqn. (1) that it is the trend (primarily) due to human-induced hydrological change. Also, because dry season loss is a variable in this regression, it is important that the authors clarify exactly which change B is tracking. As described in lines 27-28, p. 7, the dry season loss term is actually the number of dry season days, rather than a volumetric water loss. As such, the trend B presumably includes year-to-year variations in dry season water use as well. This should be stated explicitly, and instead of loss (L) in Eqn. 1, the authors should refer to the variable as what it is, number of dry season days. In summary, the manuscript needs to be more explicit about what exactly the authors intend B to include and exclude, and why.

- We thank the reviewer for these helpful comments, and particularly the suggestion to frame the response to these issues in terms of the effects on the "meaning" of the trend term B. Many of the issues relating to human-induced change (rather than environmental change drivers) were addressed through a hypothesis testing approach in a previous paper (Srinivasan et al. (2015)), which concluded that hydrologic change in the Arkavathy derives from human activity rather than changes in climate or weather.

- We agree that the designation of L as a "loss" term is misleading (as L is the time variable and rather the loss rates arise in the coefficient C,3k). We will consider changing the letter designation of the variable as well as its name in order to clarify the interpretation of that component of the regression.
- We also agree with the reviewer that the magnitude of B could be affected by other sources of variation (that is we are potentially vulnerable to the unobserved variable problem). We note that random interannual variations in water use, dry season losses, evaporative rates etc would not alter the magnitude of B, as they would not change the long-term trend. Rather, random variability would widen the confidence intervals around B. We will clarify this in the manuscript highlighting the fact that temporal changes in B that are not statistically significant may not reflect true hydrologic changes.
- Finally, we will clarify in the manuscript the statistical and hydrological interpretation of B. Statistically, B is the temporal trend in total tank water storage over time, after controlling for a stationary relationship between the covariates we describe (Ptotal, Pextreme, L) and tank water storage. Hydrologically, B represents a change in the relationship between both precipitation and dry season water losses and streamflow. Because there is is no change in the effect of dry season water losses in 6/8 watersheds, we interpret B as a change in the rainfall-runoff response. In the two subwatersheds where we detect a change in the effect of dry season water loss on tank storage, we will clarify that B captures the combined effect of hydrological change (streamflow decline pushes B in the negative direction) and dry-season tank water losses (lower tank losses pushes B in a the positive direction). Because B is negative in this area, the effect of hydrological change must exceed that of reduced tank water losses.

Secondary Concerns

The one figure that, to me, is basically a throw away is Fig. 6 for multiple reasons. First, the reservoirs are explicitly exclude from all other parts of the analysis, so whether or not their time-trends are correct is immaterial. Second, the figure does not show an independent source of the temporal evolution of reservoir extent. Third, the conclusion that can be drawn from the satellite imagery matching the timing of reservoir construction is simply that the algorithm can distinguish if, in a very large body of water, there is essentially no water or a lot of it. If this were not the case, there would be no merit in even pursuing this approach at all. It would be reasonable to mention that the method shows the timing of reservoir construction and filling as a single sentence.

- Thank you for this suggestion. We agree with the referee's argument. We will move Fig 6 to the supplementary material, and take the referee's suggestion to summarize the results in a sentence or two.

In terms of reproducibility, it would be helpful if the authors could provide contact information (an address, perhaps) for Karnataka State Remote Sensing Application Centre as a source for a shapefile of tank boundaries in the Acknowledgments section.

- We will provide contact information to KSNDMC. To assist with reproducibility, we will publish the time series of tank water area for all tanks in the watershed, along with the geolocation of each tank. This will also allow other researchers to explore the remote sensing data.

MINOR STUFF:

- Broadly we agree with all minor suggestions made by the referee and will make appropriate changes. We offer explanations below as needed.

p. 7, line 20: please clarify why average depth is used for extreme precipitation events rather than total number of extreme events or total depth of precipitation in extreme events.

- We use average depth of extreme events as a way of approximating heavy rainfall, because our experience in the field suggests a prevalence of infiltration excess runoff. Larger storms are likely to have more infiltration excess runoff due to intense rainfall, and average storm depth is a rough way of approximating this in a way that is feasible (as only daily precipitation data are available) and meets the requirements for the statistical model. Total depth in extreme events is more likely to be correlated with total precipitation depth (and thus add less information to the model) than average storm depth. We will clarify this in the paper.

p. 7, last paragraph: reference Fig. S8.

p. 8, 2nd paragraph: define variable terms explicitly (i.e., The covariates total precipitation, Ptotal,ij, : : :) here, close to the equation, instead of in previous paragraphs. State near the equation that the loss is actually the number of dry season days

p. 8 line 19: clarify what is meant by "centered" (long-term means removed?).
- We will clarify that "centered" entails removing the mean (shifting the data to a mean of zero).

Fig. 7: it would be useful to overlay a drainage/stream map to show how subwatersheds relate.

p. 10, line 1: clarify what is meant by "The spatial scales of tank clusters are comparable with that of land use"
- This sentence will be removed after making changes in the discussion. We will clarify what we originally intended to say, which is that spatial heterogeneity of hydrological change is important, and that the observed pattern of hydrological change can be related to observed pattern of land use change if we can resolve the hydrological change at a sufficient level of detail.

p. 10, lines 16-17: quotes around "drying" make sense because this is referencing algae blooms giving the false appearance of smaller tank water extent. Quotes around "wetting" do not make sense because the increase in impervious surfaces actually causes tank water extent to increase. It may not be more water in the watershed, but it is more water in the tanks.

p. 10, line 29: instead of saying ": : :by focusing on land use from a single date.", say ": : :because we only consider land use on [Mon. Day, Year]"

Figs. S4-S5: at least mention in the caption the water extent vs. precipitation plots.

References

de Graaf, I.E.M., L. P. H. van Beek, Y. Wada, M. F. P. Bierkens, 2014: Dynamic attribution of global water demand to surface water and groundwater resources: Effects of abstractions and return flows on river discharges, Adv. Water Resour., 64, 21-33, doi:10.1016/j.advwatres.2013.12.002.

---

## Author Comment (AC2) · 4 Feb 2017

*Spatial characterization of long-term hydrological change in the Arkavathy watershed adjacent to Bangalore, India -- Response to Referee 2*

**Response to Referee 2**
Referee comments in black
Our (author) responses in blue

Overall this is a well written manuscript that attempted to describe trends and spatial differences in changes in hydrology in the Arkavathy watershed on the basis of changes in extracted tank water surface area from satellite images along with other attributes.

- We thank the referee for consideration of our manuscript and valuable advice in helping us clarify some of the key messages of the paper. The referee's feedback has been helpful in alerting us to pieces of writing that need to be improved, particularly in clarifying the broader perspective.

Although the methods were well described, the broader perspective of the analysis is not well presented. After all the study analyzed the tank's surface water dynamics for a very small area (the total area of the Arkavathy is not provided), so, what new information does the findings bring to the community compared to the known facts at regional to national scale for India?

- We will provide the watershed area (4,160 sq. km) in Study Site section. We will clarify the broader implications of our research in the manuscript by making the following argument:
- The Arkavathy contains features that are characteristic of the landscape throughout much of Southern India, and although the findings from our study cannot be directly applied to the region as a whole (given the spatial heterogeneity of the change), the lessons from the Arkavathy can provide clues to hydrologic functioning in the broader region. India faces an array of water scarcity challenges, many of which have been studied at the country scale (Devineni et al., 2013; Tiwari et al., 2009) or at the local field scale (Perrin et al., 2012, Van Meter et al. 2016). Other studies have modeled hydrology at the local scale (Glendenning and Vervoort, 2011) and regional scale (Gosain et al., 2006), but none of these studies describe patterns of surface hydrological change. What is missing from the hydrology literature is an historical analysis at spatial and temporal scales commensurate with the scales of the change. The absence of hydrological records is a primary reason for this gap in the literature (Batchelor et al., 2002; Glendenning et al., 2015), and new datasets are needed that indicate hydrological change at a scale that sufficiently captures the spatial heterogeneity. Such a spatial understanding is particularly pertinent to our study region where the hydrology is truly local, because upstream and downstream subcatchments have been isolated by the fragmentation of the river network (due to tanks and check dams) and the subsurface disconnection due to the vastly depleted groundwater table (as we will clarify in our manuscript, urban effluent can serve to maintain a connected river network directly downstream of urban areas). The heterogeneity of observed changes in the Arkavathy emphasizes one of the problems associated with viewing water trends only at regional or national levels - such large scale trends to not map directly to local scales, yet these are the scales at which people experience and must respond to change. Such local understanding is of great importance to water managers in southern India, as

considerable efforts are underway for river and tank rehabilitation in some areas, without a clear understanding of the mechanisms underlying the historical degradation and loss of water resources (Kumar et al., 2016; Srinivasan et al., 2014).

Given the size of the tanks studied, I would imagine the seasonal water area dynamics will have greater implications than the inter-annual dynamics. The manuscript did not discuss anything on the seasonality for these tanks, or how does that influence the trend?

- We agree that seasonal dynamics are interesting to understand in so far as they indicate the seasonal availability of surface water resources. However, we avoided a detailed description of these dynamics for several reasons. Firstly, since tanks are not widely utilized as a surface water resource throughout the Arkavathy Basin today, the importance of understanding these seasonal dynamics is not so great in the present context as in situations where those surface water stores are relied upon by communities. The importance of the tanks as studied in this paper is as indicators of long-term changes through space in the hydrological dynamics that produce the end of monsoon season storage. Secondly, for pragmatic reasons, it is challenging to study within-year variations other than in the dry season. For approximately 6 months of the year, extensive cloud cover obscures many of the tanks in Landsat images and active radar satellite imagery (which can effectively "see through" clouds) is too coarse to estimate water area in small tanks. We appreciate the referee comments and we will more carefully discuss dry-season dynamics in the manuscript.

The manuscript mentioned about differences in water quality, turbidity, vegetation in the water which are influential factors for changes in the reflectance. Even though the DN values were converted to reflectance, the manuscript used only one index (NDWI) to classify water surface area, while there were potentially many other methods or index (Senay et al., 2013) could be used to map water surface correctly, as no one index can cover it all.

- We agree that there is no one method for remote sensing classification of surface water. We selected a simple classification method that was consistent across all Landsat sensors (MSS, TM, ETM, OLI). Our method uses NDWI as an initial classification, and we then apply spectral unmixing using Red, Green, and NIR bands. Although more complex methods have been published, they may not result in a significant improvement in confidence in our model, which we believe is sufficient for our purposes.

While the analysis was performed for the time period between 1972 and 2010 the validation was done for 2014 results. To me validation needs to be done for the time for which the trend analysis is performed (few sample years both wet and dry between 1972 and 2010).

As the study area is so small Google earth might provide good data for validation. Have the authors looked into google earth images as a potential source of validation data?

- We thank the referee for this suggestion. Our ability to completely validate the model between 1972 and 2010 is limited by the availability of independent data-sources at higher spatial resolution for such a validation - specifically, the lack of accessible aerial

data for the region and the lack of low-cloud commercial high-resolution satellite datasets prior to early 2000s. Since, however, there is no reason to anticipate that the classification relationships should be non-stationary, we consider the most compelling part of the reviewer's suggestion is to address both relatively wet and relatively dry years, which can be accomplished using a more contemporary dataset. In particular, the suggestion to use Digital Globe (DG) images via Google Earth is sensible, and allows us to use images from as early as 2004 (although we note that individual DG images cover only a portion of the whole Arkavathy, so that the earliest date of available imagery varies). Specifically, there are DG images which may be suitable for validation (being close to the end of the monsoon season and having a suitable Landsat image taken at a similar time) and covering portions of the watershed on the following dates : 7-Dec-2005, 30-Dec-2006, 30-Dec-2007, and 25-Feb-2009, 7-Feb-2004 On 11-Feb-2009, and 8-Feb-2010.

- We are working on the details of a validation approach based on manual delineation of tank water area from the DG imagery which will be included in the revised paper. The scale of the validation in terms of the minimal number of tanks required will be decided via power analysis as follows: We set the null hypotheses that the actual correlation between the area of classified tanks and the area of validation tanks is greater than the correlation for tanks classified in our initial (2014) analysis described in the manuscript (H0: $R^2 > 0.95$). The null hypothesis is therefore that the actual $R^2$ is less than 0.95. If the true $R^2$ is 0.9, we would need 30-50 tanks to achieve a power of 0.5-0.75 in this statistical test to reject the null hypothesis. We will attempt to reach this number in multiple years, noting the limited spatial scale of DG images and limited date range (2004 and later).

Page 10 line 5: claims that MK analysis confirms an increase in agricultural land use fraction is related to decrease in tank water storage. How? There is no evidence shown in the manuscript that suggests agricultural land use is increasing. This is vague to me.

- We are going to restructure this analysis, and will make sure to clarify a number of key points. Agriculture has not expanded so much as it has changed over the course of the study period, and the changes in the nature of agriculture could be the cause of drying in the norther part of the Arkavathy. Bangalore has urbanized rapidly over the study period, with its population increasing by a factor of 4. We will clarify these points in the revised manuscript. Furthermore, using recently developed land use maps from 1973, 1994, 2001, and 2013 of the northern Arkavathy watershed (the 3 northernmost subcatchments), we will explore a more detailed analysis of the relationship between land use and hydrological change and present any additional findings in the updated manuscript.

Page 10 line 11-12: statement connects with changes in land use and management practice with depleted subsurface stores without providing evidence.

- We will restructure this analysis as well. We will provide more context regarding changes in land use as well as management practices. Our discussion was intended as an initial

attempt as understanding drivers of hydrological change. We will clarify that this analysis is exploratory, and we will also provide more details from other works that have been written already (Srinivasan et al., 2015; Lele et al., 2014).

Page 11 line 6-7: Target for classification is to identify water and not water cells, in that case how does incorporation of additional land cover will reduce the classification error?
- Because we are using spectral unmixing, the land class end-member affects the calculated water fraction in each cell. For this reason, having additional (and more precise land classes) could potentially improve classification. We will clarify this point further in the paper.

I think the method used in the manuscript is too simplistic, although producing time- series information of tank water surface area is valuable. I am not sure how much new information has been brought to the community by this study; therefore I am not convinced that HESS is the right journal for this article.
- We agree that the classification is fairly simple, but overcomes a variety of challenges related to the study, such as the need to incorporate imagery from four Landsat sensors (MSS, TM, ETM, OLI), spectral unmixing in all images, cloud and cloud shadow masking, and the temporal nature of water in tanks (and single image gap filling in SLF-off images). We also note that the classification serves its purpose based on the validation we showed in the manuscript. The overall objectives for the paper (and updated validation information) will be clarified in the updated manuscript, as we describe above and in the letter to the editor.

Senay, G.B., Velpuri, N.M., Henok, A., Pervez, M.S., Asante, K.O., Gatarwa, K., Asefa, T., & Jay, A. (2013). Establishing an operational waterhole monitoring system using satellite data and hydrologic modelling: Application in the pastoral regions of East Africa. Pastoralism: Research, Policy and Practice, 3, 20.

---

## Author Comment (AC3) · 4 Feb 2017

***Spatial characterization of long-term hydrological change in the Arkavathy watershed adjacent to Bangalore, India -- Response to Editor and Anonymous Referees 1 & 2***

Dear Dr. Shraddhanand Shukla,

Thank you for conveying the referee comments to us. Both referees provided insightful comments that we believe will strengthen our manuscript. In response to their comments, we intend to make several revisions, including additional validation of our classification method and a modified discussion of the relationship between hydrological change and land use. We outline our major edits and additions in this letter. Other revisions and responses to more minor comments are included in the attached detailed responses to the referees.

Referee 1 raised two main concerns. The first related to (a) our ability to attribute hydrological change to land use and the second to (b) our treatment of dry-season water losses. With respect to point (a), we agree with the referee that we did not present a detailed evidence-base that quantifies the role of land use and land change in shaping hydrological change in the Arkavathy. Such a full attribution lies beyond the scope of the present paper, and forms the subject of an ongoing, separate study. We also agree that we need to offer more information regarding the feasible drivers of the observed hydrologic change (a topic that we explored in greater detail in a previous paper, Srinivasan et al., 2015) as context for the present study.

Therefore, we propose removing Figure 8b from the manuscript, and rewriting the discussion about potential drivers of hydrologic change that summarizes the available evidence (including that presented in Srinivasan et al., 2015) and provides an open-ended discussion of the spatial trends observed, and their interpretation in light of spatial patterns of change (including changes in land use and land use practices) and hydrological processes occurring in the basin. We will also include the possibility that the upstream-downstream dynamic could play a role in the pattern of hydrological changes, but our analysis to this point suggests that it is unlikely because of the overall watershed fragmentation and disconnection of surface water and groundwater (except downstream of urban areas which are affected by inter-basin water imports and urban effluent). Lastly, we will explore a more sophisticated analysis of hydrological trends and landuse in the northern part of the watershed and will present any additional findings in the revised manuscript.

With respect to point (b), we thank the reviewer for pointing out the potential difficulties in interpretation posed by the fact that the dry-season water loss metric used in the overall statistical model did not allow for that metric to vary over time, yet independent exploration of losses in the two northernmost watersheds suggested that the loss rate was time varying. While we agree that this situation is potentially problematic, we note that the direction of the trend in losses (decreasing over time) is opposed to the overall trend in drying in the watersheds. In other words, the assumption of stationarity in these 2 watersheds is a conservative assumption that will lead to us under-estimating the time trend in drying, and does not change the conclusions obtained by the statistical model. We will clarify this issue in the revised manuscript.

Referee 1 also made comments regarding our figures. In response to his suggestions, we will move Fig. 6 to the Supplementary Material, and Figs. S4 and S5 to the results section,

excluding the comparison of water extent and precipitation which will remain in the Supplementary Material.

Referee 2 raised several significant high level concerns. In particular, they questioned (a) the suitability of the manuscript for HESS, (b) what relevant conclusions about regional and national water resources could be drawn from understanding the relatively small spatial scales addressed within the Arkavathy Basin, (c) the simplicity of the remote sensing classification, (d) the suitability of validation of the remote sensing classification, and (e) failure to address seasonal dynamics in the study.

To address (a) and (b), we will restructure the introduction and discussion sections highlighting the following arguments to emphasize the significance of this paper to the water resources community in southern India as well as the hydrology research community. Our revised introduction will make the following points:

India faces an array of water scarcity challenges, many of which have been studied at the country scale (Devineni et al., 2013; Tiwari et al., 2009) or at the local field scale (Perrin et al., 2012, Van Meter et al. 2016). Other studies have applied hydrological models at the local scale (Glendenning and Vervoort, 2011) and regional scale (Gosain et al., 2006), but none of these studies describe patterns of surface hydrological change. What is missing from the hydrology literature is an historical analysis at spatial and temporal scales commensurate with the scales of the change. The absence of hydrological records is a primary reason for this gap in the literature (Batchelor et al., 2003; Glendenning et al., 2015), and new datasets are needed that indicate hydrological change at a scale that sufficiently captures the spatial heterogeneity. A study in Tamil Nadu considered changes in tank water storage at multiple points in time (Mialhe et al 2008), but to our knowledge there are no other studies that identify distributed hydrological changes throughout space. Such a spatial understanding is particularly pertinent to our study region where the hydrology is truly local. Hydrological records are insufficient to capture the spatial nature of hydrological change, as there are only two streamflow gauges in the Arkavathy watershed which spans over 4,000 sq. km. Furthermore, streamflow in the Arkavathy at a given point is not an integrated measure of upstream processes. Upstream and downstream subcatchments have been isolated by the fragmentation of the river network (due to tanks and check dams) and the subsurface disconnection due to the vastly depleted groundwater table (as we will clarify in our manuscript, urban effluent can serve to maintain a connected river network directly downstream of urban areas). The Arkavathy contains features that are characteristic of the landscape throughout much of Southern India, and although the findings from our study cannot be directly applied to the region as a whole (given the spatial heterogeneity of the change), the lessons from the Arkavathy can provide clues to hydrologic functioning in the broader region. The heterogeneity of observed changes in the Arkavathy emphasizes one of the problems associated with viewing water trends only at regional or national levels -- such large scale trends to not map directly to local scales, yet these are the scales at which people experience and must respond to change. Such local understanding is of great importance to water managers in southern India, as considerable efforts are underway for river and tank

rehabilitation in some areas, without a clear understanding of the mechanisms underlying the historical degradation and loss of water resources (Kumar et al., 2016; Srinivasan et al., 2015).

Regarding point (c), we agree that the remote-sensing classification is simple, but that it has strengths: it is likely to be unbiased and stationary across all Landsat sensors and furthermore, we have demonstrated that it is sufficiently accurate for our purposes. We also note that although the reviewer suggested that our classification was solely based on NDWI, that NDWI in fact represents only the first stage of the classification, which then relies on use of the green, red, and NIR bands in our spectral unmixing of clear and turbid water.

We appreciated the referees suggestions regarding point (d). Google Earth images do indeed offer a high resolution visual dataset that gives us the opportunity to further validate the remote sensing approach. We plan to incorporate such additional validation in the revised manuscript. The earliest Google Earth images in our study watershed are in 2004. The details of our proposed additional validation strategy are outlined in the detailed response to Reviewer 2.

Finally, with respect to point (e), we agree that seasonal dynamics are interesting to understand in so far as they indicate the seasonal availability of surface water resources. However, we avoided a detailed description of these dynamics for several reasons. Firstly, since tanks are not widely utilized as a surface water resource throughout the Arkavathy Basin today, the importance of understanding these seasonal dynamics is not so great in the present context as in situations where those surface water stores are relied upon by communities. The importance of the tanks as studied in this paper is as indicators of long-term changes through space in the hydrological dynamics that produce the end of monsoon season storage. Secondly, for pragmatic reasons, it is challenging to study within-year variations other than in the dry season. For approximately six months of the year, extensive cloud cover obscures many of the tanks in Landsat images and active radar satellite imagery (which can effectively "see through" clouds) is too coarse to estimate water area in small tanks. We appreciate the referee comments and we will more carefully discuss dry-season dynamics in the manuscript.

Both referees provided clear suggestions for improving the manuscript. We thank the referees for their consideration of this paper and we include detailed responses to the referees below. We look forward to your comments on the manuscript and our response to the referees.

Best regards,

Gopal Penny
Veena Srinivasan
Iryna Dronova
Sharad Lele
Sally Thompson

**References**

Batchelor, C., Rama Mohan Rao, M., & Manohar Rao, S. (2003). Watershed development: A solution to water shortages in semi-arid India or part of the problem. *Land Use and Water Resources Research*, 1–10.

Devineni, N., Perveen, S., & Lall, U. (2013). Assessing chronic and climate-induced water risk through spatially distributed cumulative deficit measures: A new picture of water sustainability in India. *Water Resources Research*, *49*(4), 2135–2145. http://doi.org/10.1002/wrcr.20184

Glendenning, C. J., van Ogtrop, F. F., Mishra, a. K., & Vervoort, R. W. (2012). Balancing watershed and local scale impacts of rain water harvesting in India—A review. *Agricultural Water Management*, *107*, 1–13. http://doi.org/10.1016/j.agwat.2012.01.011

Glendenning, C. J., & Vervoort, R. W. (2011). Hydrological impacts of rainwater harvesting (RWH) in a case study catchment: The Arvari River, Rajasthan, India. *Agricultural Water Management*, *98*(4), 715–730. http://doi.org/10.1016/j.agwat.2010.11.010

Gosain, A., Rao, S., & Basuray, D. (2006). Climate change impact assessment on hydrology of Indian river basins. *Current Science*, *90*(3). Retrieved from http://www.iisc.ernet.in/currsci/feb102006/346.pdf

Kumar, M. D., Bassi, N., Kishan, K. S., Chattopadhyay, S., & Ganguly, A. (2016). Rejuvenating Tanks in Telangana. *Economic & Political Weekly*, *lI*(34), 30–34.

Mialhe, F., Gunnell, Y., & Mering, C. (2008). Synoptic assessment of water resource variability in reservoirs by remote sensing: General approach and application to the runoff harvesting systems of south India. *Water Resources Research*, *44*(5), n/a-n/a. http://doi.org/10.1029/2007WR006065

Perrin, J., Ferrant, S., Massuel, S., Dewandel, B., Maréchal, J. C., Aulong, S., & Ahmed, S. (2012). Assessing water availability in a semi-arid watershed of southern India using a semi-distributed model. *Journal of Hydrology*, *460–461*, 143–155. http://doi.org/10.1016/j.jhydrol.2012.07.002

Srinivasan, V., Thompson, S., Madhyastha, K., Penny, G., Jeremiah, K., & Lele, S. (2015). Why is the Arkavathy River drying? A multiple hypothesis approach in a data scarce region. *Hydrology and Earth System Sciences Discussions*, *12*, 25–66. http://doi.org/10.5194/hessd-12-25-2015

Tiwari, V. M., Wahr, J., & Swenson, S. (2009). Dwindling groundwater resources in northern India, from satellite gravity observations. *Geophysical Research Letters*, *36*(18), L18401. http://doi.org/10.1029/2009GL039401

Van Meter, K. J., Basu, N. B., McLaughlin, D. L., & Steiff, M. (2016). The socio-ecohydrology of rainwater harvesting in India: understanding water storage and release dynamics at tank and catchment scales. *Hydrology and Earth System Sciences Discussions*, *20*, 2629–2647. http://doi.org/10.5194/hessd-12-12121-2015

**Response to Referee 1**
Referee comments in black
Our (author) response in blue

Strengths

The premise of this paper is interesting and the application of remote sensing to measuring the extent of tank surface area is unique. The paper demonstrates the practical application of remote sensing for characterizing hydrologic change in an otherwise unmonitored setting.

I appreciate the challenge of accounting for various degrees of turbidity in the classification of these water bodies. The methods for measuring tank water extent were clearly presented, and the supplementary figures showing examples of classification were really useful. In my opinion, all of the figures and tables, including those in the supplement, are necessary and contribute to this paper, with the possible exception of Fig. 6. The supplemental tables should make it possible for someone to reproduce this
Analysis.

- We thank the referee for offering careful consideration and analysis of the manuscript. The referee brings up a number of valid points that we believe will strengthen the paper. We intend to incorporate a number of the suggestions from this referee. We will move Figure 6 to the supplementary material. We will move figures S4 and S5 from the supplementary material to the Results section of the manuscript (excluding the comparisons of water extent and precipitation, which will remain in the supplementary material).

Major Concerns

Although I accept that the multiple regression in Eqn. 1 is a reasonable technique to remove precipitation (climate) effects from the estimate of long-term trend, the analysis of hydrologic change related to land use change is not convincing. The visual comparison of percent agriculture with temporal trend in water extent shown in Fig. 8b does not show a clear relationship. It appears that there is only a temporal trend of magnitude greater than 1 ha decadeˆ(-1) 10 kmˆ(-2) (units should be clarified, is this ha/(decade * 10 kmˆ2)?) if the agricultural area is close to 0.75% (which I assume is a typo for 75%); however, low temporal trends are possible for any percent of agricultural land area. This is not a strong argument for a relationship between the two. In fact, the notable negative trends occur only in the two northernmost sub-catchments.

- Thank you for this comment. Our intention in presenting Figure 8b was to conclude the manuscript with some initial ideas about the attribution of hydrologic change to potential drivers. We agree, however, that despite the statistically significant Mann-Kendall trend, the proportion of a watershed covered with agricultural land use has a tenuous

relationship with hydrological change (at a minimum, one might argue that using a snapshot of land use to explain a decadal trend in hydrology poses a problematic mismatch). We agree that removing Figure 8b is appropriate. We will replace it with a written discussion that offers a broader context for the observed changes. Furthermore, using recently developed maps of land use from 1973, 1994, 2001, and 2013 of the northern Arkavathy watershed (the 3 northernmost subcatchments), we will explore a more detailed analysis of the relationship between land use and hydrological change and present any additional findings in the updated manuscript.

An argument could possibly be made that this is an upstream-to-downstream effect, where water withdrawals upstream have a greater impact on stored water over time because return flows from irrigation dampen the effects of water withdrawals in downstream sub-catchments and/or the major reservoirs shown on Fig. 1 are operated in a way that mitigates long-term trends in water storage changes in the tanks (see for example de Graaf et al., 2014).

- Thank you for this interesting suggestion. Our current hypothesis is that the drying of the northern part of the watershed is linked to groundwater pumping that caused a disconnection between groundwater and surface water (see Srinivasan et al., 2015), leading to reduced baseflow, in a manner analogous to the model in de Graaf et al. (2014). Field studies addressing this mechanism are also currently in progress. Unlike the model presented by de Graaf, however, we are doubtful that the upstream-to-downstream effect is important in the Arkavathy today. Various sources of indirect evidence indicate that the water table is hundreds of meters below the surface in northern parts of the Arkavathy watershed (Srinivasan et al., 2015), suggesting that excess infiltration water is likely to move vertically. Similarly, the relief in the watershed is only about 100 m over a distance of 100 km, again promoting vertical groundwater movement and system-wide return flows connecting upstream to downstream are unlikely. We will make a note to this effect in the revised manuscript.

Additionally, as the authors note on p. 7, lines 28-35, the two watersheds farthest upstream (those that drive the trend) were the only two watersheds with a significant trend in dry season water loss, which they relate to the shift from tank irrigation to groundwater irrigation during the study period. Unless I have misunderstood how dry season losses were treated in the regression, this shift would be reflected in the long-term trend. The authors should test whether or not the change in drying rate is the dominant cause of the trend, and if without this shift, a relationship with the % agricultural area still holds.

- Thank you for this salient observation. Non-stationarity in the dry-season water loss term would indeed affect the magnitude of estimated hydrological change in tank clusters given that the regression relationship used to identify this change assumes a stationary loss coefficient.
- The violation of this stationarity assumption in 2 tank clusters might be expected to marginally increase the model error, and, if the time trend in the dry season losses was

aligned with the time trend in tank storage, it could indeed confound interpretation of the meaning of the storage trend. However, the trend in dry season losses in the northernmost tank clusters is, instead, in the opposite direction to the trend in storage. Dry-season loss rates have *decreased* over time in the two northernmost subwatersheds. We would expect this change to result in an *increase* in tank water storage after monsoon season (as tanks lose water more slowly). Yet we observe a statistically significant decrease in post-monsoon tank storage over time, in spite of the decrease in loss rates. Thus, introducing a non-stationary loss coefficient into the model might improve model fit (at the expense of the degrees of freedom of the model) and improve quantitative estimates of the rate of drying due to hydrologic change in the northern watersheds, but would not alter the main conclusion of the study, which is that these watersheds are, in fact, drying.
- We will add clarification of these points to the discussion.

Are there other spatial patterns in rates of groundwater pumping?
- Understanding the spatial patterns of groundwater extraction in the Arkavathy Basin would be very useful. Unfortunately, monitoring of groundwater use through space has been indirect and sparse. We are exploring whether proxies for groundwater irrigation could be developed from the remote sensing record as an ongoing project, but at this point we are not in a strong position to analyze the effects of such spatial variation on the spatial differences in surface water trends.

The authors develop a simple mathematical model to extract the trend (B) due to "hydrological change", by which I infer that the authors are referring to the "temporal trends in water extent: : :indicative of long-tem hydrological changes induced by human activity" (p. 3, lines 12-13). The intent would be clearer if the authors were to describe other potential causes of this change (for example, temperature change in the region) and to state when defining B in Eqn. (1) that it is the trend (primarily) due to human-induced hydrological change. Also, because dry season loss is a variable in this regression, it is important that the authors clarify exactly which change B is tracking. As described in lines 27-28, p. 7, the dry season loss term is actually the number of dry season days, rather than a volumetric water loss. As such, the trend B presumably includes year-to-year variations in dry season water use as well. This should be stated explicitly, and instead of loss (L) in Eqn. 1, the authors should refer to the variable as what it is, number of dry season days. In summary, the manuscript needs to be more explicit about what exactly the authors intend B to include and exclude, and why.

- We thank the reviewer for these helpful comments, and particularly the suggestion to frame the response to these issues in terms of the effects on the "meaning" of the trend term B. Many of the issues relating to human-induced change (rather than environmental change drivers) were addressed through a hypothesis testing approach in a previous paper (Srinivasan et al. (2015)), which concluded that hydrologic change in the Arkavathy derives from human activity rather than changes in climate or weather.

- We agree that the designation of L as a "loss" term is misleading (as L is the time variable and rather the loss rates arise in the coefficient C,3k). We will consider changing the letter designation of the variable as well as its name in order to clarify the interpretation of that component of the regression.
- We also agree with the reviewer that the magnitude of B could be affected by other sources of variation (that is we are potentially vulnerable to the unobserved variable problem). We note that random interannual variations in water use, dry season losses, evaporative rates etc would not alter the magnitude of B, as they would not change the long-term trend. Rather, random variability would widen the confidence intervals around B. We will clarify this in the manuscript highlighting the fact that temporal changes in B that are not statistically significant may not reflect true hydrologic changes.
- Finally, we will clarify in the manuscript the statistical and hydrological interpretation of B. Statistically, B is the temporal trend in total tank water storage over time, after controlling for a stationary relationship between the covariates we describe (Ptotal, Pextreme, L) and tank water storage. Hydrologically, B represents a change in the relationship between both precipitation and dry season water losses and streamflow. Because there is is no change in the effect of dry season water losses in 6/8 watersheds, we interpret B as a change in the rainfall-runoff response. In the two subwatersheds where we detect a change in the effect of dry season water loss on tank storage, we will clarify that B captures the combined effect of hydrological change (streamflow decline pushes B in the negative direction) and dry-season tank water losses (lower tank losses pushes B in a the positive direction). Because B is negative in this area, the effect of hydrological change must exceed that of reduced tank water losses.

Secondary Concerns

The one figure that, to me, is basically a throw away is Fig. 6 for multiple reasons. First, the reservoirs are explicitly exclude from all other parts of the analysis, so whether or not their time-trends are correct is immaterial. Second, the figure does not show an independent source of the temporal evolution of reservoir extent. Third, the conclusion that can be drawn from the satellite imagery matching the timing of reservoir construction is simply that the algorithm can distinguish if, in a very large body of water, there is essentially no water or a lot of it. If this were not the case, there would be no merit in even pursuing this approach at all. It would be reasonable to mention that the method shows the timing of reservoir construction and filling as a single sentence.

- Thank you for this suggestion. We agree with the referee's argument. We will move Fig 6 to the supplementary material, and take the referee's suggestion to summarize the results in a sentence or two.

In terms of reproducibility, it would be helpful if the authors could provide contact information (an address, perhaps) for Karnataka State Remote Sensing Application Centre as a source for a shapefile of tank boundaries in the Acknowledgments section.

- We will provide contact information to KSNDMC. To assist with reproducibility, we will publish the time series of tank water area for all tanks in the watershed, along with the geolocation of each tank. This will also allow other researchers to explore the remote sensing data.

MINOR STUFF:

- Broadly we agree with all minor suggestions made by the referee and will make appropriate changes. We offer explanations below as needed.

p. 7, line 20: please clarify why average depth is used for extreme precipitation events rather than total number of extreme events or total depth of precipitation in extreme events.

- We use average depth of extreme events as a way of approximating heavy rainfall, because our experience in the field suggests a prevalence of infiltration excess runoff. Larger storms are likely to have more infiltration excess runoff due to intense rainfall, and average storm depth is a rough way of approximating this in a way that is feasible (as only daily precipitation data are available) and meets the requirements for the statistical model. Total depth in extreme events is more likely to be correlated with total precipitation depth (and thus add less information to the model) than average storm depth. We will clarify this in the paper.

p. 7, last paragraph: reference Fig. S8.

p. 8, 2nd paragraph: define variable terms explicitly (i.e., The covariates total precipitation, Ptotal,ij, : : :) here, close to the equation, instead of in previous paragraphs. State near the equation that the loss is actually the number of dry season days

p. 8 line 19: clarify what is meant by "centered" (long-term means removed?).
- We will clarify that "centered" entails removing the mean (shifting the data to a mean of zero).

Fig. 7: it would be useful to overlay a drainage/stream map to show how subwatersheds relate.

p. 10, line 1: clarify what is meant by "The spatial scales of tank clusters are comparable with that of land use"
- This sentence will be removed after making changes in the discussion. We will clarify what we originally intended to say, which is that spatial heterogeneity of hydrological change is important, and that the observed pattern of hydrological change can be related to observed pattern of land use change if we can resolve the hydrological change at a sufficient level of detail.

p. 10, lines 16-17: quotes around "drying" make sense because this is referencing algae blooms giving the false appearance of smaller tank water extent. Quotes around "wetting" do not make sense because the increase in impervious surfaces actually causes tank water extent to increase. It may not be more water in the watershed, but it is more water in the tanks.

p. 10, line 29: instead of saying ": : :by focusing on land use from a single date.", say ": : :because we only consider land use on [Mon. Day, Year]"

Figs. S4-S5: at least mention in the caption the water extent vs. precipitation plots.

References

de Graaf, I.E.M., L. P. H. van Beek, Y. Wada, M. F. P. Bierkens, 2014: Dynamic attribution of global water demand to surface water and groundwater resources: Effects of abstractions and return flows on river discharges, Adv. Water Resour., 64, 21-33, doi:10.1016/j.advwatres.2013.12.002.

**Response to Referee 2**

Referee comments in black

Our (author) responses in blue

Overall this is a well written manuscript that attempted to describe trends and spatial differences in changes in hydrology in the Arkavathy watershed on the basis of changes in extracted tank water surface area from satellite images along with other attributes.

- We thank the referee for consideration of our manuscript and valuable advice in helping us clarify some of the key messages of the paper. The referee's feedback has been helpful in alerting us to pieces of writing that need to be improved, particularly in clarifying the broader perspective.

Although the methods were well described, the broader perspective of the analysis is not well presented. After all the study analyzed the tank's surface water dynamics for a very small area (the total area of the Arkavathy is not provided), so, what new information does the findings bring to the community compared to the known facts at regional to national scale for India?

- We will provide the watershed area (4,160 sq. km) in Study Site section. We will clarify the broader implications of our research in the manuscript by making the following argument:
- The Arkavathy contains features that are characteristic of the landscape throughout much of Southern India, and although the findings from our study cannot be directly applied to the region as a whole (given the spatial heterogeneity of the change), the lessons from the Arkavathy can provide clues to hydrologic functioning in the broader region. India faces an array of water scarcity challenges, many of which have been studied at the country scale (Devineni et al., 2013; Tiwari et al., 2009) or at the local field scale (Perrin et al., 2012, Van Meter et al. 2016). Other studies have modeled hydrology at the local scale (Glendenning and Vervoort, 2011) and regional scale (Gosain et al., 2006), but none of these studies describe patterns of surface hydrological change. What is missing from the hydrology literature is an historical analysis at spatial and temporal scales commensurate with the scales of the change. The absence of hydrological records is a primary reason for this gap in the literature (Batchelor et al., 2002; Glendenning et al., 2015), and new datasets are needed that indicate hydrological change at a scale that sufficiently captures the spatial heterogeneity. Such a spatial understanding is particularly pertinent to our study region where the hydrology is truly local, because upstream and downstream subcatchments have been isolated by the fragmentation of the river network (due to tanks and check dams) and the subsurface disconnection due to the vastly depleted groundwater table (as we will clarify in our manuscript, urban effluent can serve to maintain a connected river network directly downstream of urban areas). The heterogeneity of observed changes in the Arkavathy emphasizes one of the problems associated with viewing water trends only at regional or national levels - such large scale trends to not map directly to local scales, yet these are the scales at which people experience and must respond to change. Such local understanding is of great importance to water managers in southern India, as

considerable efforts are underway for river and tank rehabilitation in some areas, without a clear understanding of the mechanisms underlying the historical degradation and loss of water resources (Kumar et al., 2016; Srinivasan et al., 2014).

Given the size of the tanks studied, I would imagine the seasonal water area dynamics will have greater implications than the inter-annual dynamics. The manuscript did not discuss anything on the seasonality for these tanks, or how does that influence the trend?

- We agree that seasonal dynamics are interesting to understand in so far as they indicate the seasonal availability of surface water resources. However, we avoided a detailed description of these dynamics for several reasons. Firstly, since tanks are not widely utilized as a surface water resource throughout the Arkavathy Basin today, the importance of understanding these seasonal dynamics is not so great in the present context as in situations where those surface water stores are relied upon by communities. The importance of the tanks as studied in this paper is as indicators of long-term changes through space in the hydrological dynamics that produce the end of monsoon season storage. Secondly, for pragmatic reasons, it is challenging to study within-year variations other than in the dry season. For approximately 6 months of the year, extensive cloud cover obscures many of the tanks in Landsat images and active radar satellite imagery (which can effectively "see through" clouds) is too coarse to estimate water area in small tanks. We appreciate the referee comments and we will more carefully discuss dry-season dynamics in the manuscript.

The manuscript mentioned about differences in water quality, turbidity, vegetation in the water which are influential factors for changes in the reflectance. Even though the DN values were converted to reflectance, the manuscript used only one index (NDWI) to classify water surface area, while there were potentially many other methods or index (Senay et al., 2013) could be used to map water surface correctly, as no one index can cover it all.

- We agree that there is no one method for remote sensing classification of surface water. We selected a simple classification method that was consistent across all Landsat sensors (MSS, TM, ETM, OLI). Our method uses NDWI as an initial classification, and we then apply spectral unmixing using Red, Green, and NIR bands. Although more complex methods have been published, they may not result in a significant improvement in confidence in our model, which we believe is sufficient for our purposes.

While the analysis was performed for the time period between 1972 and 2010 the validation was done for 2014 results. To me validation needs to be done for the time for which the trend analysis is performed (few sample years both wet and dry between 1972 and 2010).

As the study area is so small Google earth might provide good data for validation. Have the authors looked into google earth images as a potential source of validation data?

- We thank the referee for this suggestion. Our ability to completely validate the model between 1972 and 2010 is limited by the availability of independent data-sources at higher spatial resolution for such a validation - specifically, the lack of accessible aerial

data for the region and the lack of low-cloud commercial high-resolution satellite datasets prior to early 2000s. Since, however, there is no reason to anticipate that the classification relationships should be non-stationary, we consider the most compelling part of the reviewer's suggestion is to address both relatively wet and relatively dry years, which can be accomplished using a more contemporary dataset. In particular, the suggestion to use Digital Globe (DG) images via Google Earth is sensible, and allows us to use images from as early as 2004 (although we note that individual DG images cover only a portion of the whole Arkavathy, so that the earliest date of available imagery varies). Specifically, there are DG images which may be suitable for validation (being close to the end of the monsoon season and having a suitable Landsat image taken at a similar time) and covering portions of the watershed on the following dates : 7-Dec-2005, 30-Dec-2006, 30-Dec-2007, and 25-Feb-2009, 7-Feb-2004 On 11-Feb-2009, and 8-Feb-2010.

- We are working on the details of a validation approach based on manual delineation of tank water area from the DG imagery which will be included in the revised paper. The scale of the validation in terms of the minimal number of tanks required will be decided via power analysis as follows: We set the null hypotheses that the actual correlation between the area of classified tanks and the area of validation tanks is greater than the correlation for tanks classified in our initial (2014) analysis described in the manuscript (H0: $R^2 > 0.95$). The null hypothesis is therefore that the actual $R^2$ is less than 0.95. If the true $R^2$ is 0.9, we would need 30-50 tanks to achieve a power of 0.5-0.75 in this statistical test to reject the null hypothesis. We will attempt to reach this number in multiple years, noting the limited spatial scale of DG images and limited date range (2004 and later).

Page 10 line 5: claims that MK analysis confirms an increase in agricultural land use fraction is related to decrease in tank water storage. How? There is no evidence shown in the manuscript that suggests agricultural land use is increasing. This is vague to me.

- We are going to restructure this analysis, and will make sure to clarify a number of key points. Agriculture has not expanded so much as it has changed over the course of the study period, and the changes in the nature of agriculture could be the cause of drying in the norther part of the Arkavathy. Bangalore has urbanized rapidly over the study period, with its population increasing by a factor of 4. We will clarify these points in the revised manuscript. Furthermore, using recently developed land use maps from 1973, 1994, 2001, and 2013 of the northern Arkavathy watershed (the 3 northernmost subcatchments), we will explore a more detailed analysis of the relationship between land use and hydrological change and present any additional findings in the updated manuscript.

Page 10 line 11-12: statement connects with changes in land use and management practice with depleted subsurface stores without providing evidence.

- We will restructure this analysis as well. We will provide more context regarding changes in land use as well as management practices. Our discussion was intended as an initial

attempt as understanding drivers of hydrological change. We will clarify that this analysis is exploratory, and we will also provide more details from other works that have been written already (Srinivasan et al., 2015; Lele et al., 2014).

Page 11 line 6-7: Target for classification is to identify water and not water cells, in that case how does incorporation of additional land cover will reduce the classification error?

- Because we are using spectral unmixing, the land class end-member affects the calculated water fraction in each cell. For this reason, having additional (and more precise land classes) could potentially improve classification. We will clarify this point further in the paper.

I think the method used in the manuscript is too simplistic, although producing time- series information of tank water surface area is valuable. I am not sure how much new information has been brought to the community by this study; therefore I am not convinced that HESS is the right journal for this article.

- We agree that the classification is fairly simple, but overcomes a variety of challenges related to the study, such as the need to incorporate imagery from four Landsat sensors (MSS, TM, ETM, OLI), spectral unmixing in all images, cloud and cloud shadow masking, and the temporal nature of water in tanks (and single image gap filling in SLF-off images). We also note that the classification serves its purpose based on the validation we showed in the manuscript. The overall objectives for the paper (and updated validation information) will be clarified in the updated manuscript, as we describe above and in the letter to the editor.

Senay, G.B., Velpuri, N.M., Henok, A., Pervez, M.S., Asante, K.O., Gatarwa, K., Asefa, T., & Jay, A. (2013). Establishing an operational waterhole monitoring system using satellite data and hydrologic modelling: Application in the pastoral regions of East Africa. Pastoralism: Research, Policy and Practice, 3, 20.

---

## Author Response (AR1)

***Spatial characterization of long-term hydrological change in the Arkavathy watershed adjacent to Bangalore, India -- (1) Letter to Editor, (2) list of manuscript changes, (3) responses to Referees 1 & 2, and (4) marked up version of revised manuscript***

Dear Dr. Shraddhanand Shukla,

Thank you for conveying the referee comments and major revisions to us. We made a number of improvements to the revised manuscript in response to the referee comments. We have provided a list of manuscript changes below, followed by point-by-point responses to the referee comments, and finally a marked up version of the revised manuscript with pertinent revisions colored in red.

Among the changes to the manuscript, the most notable was that we explored a more sophisticated analysis of the associating human drivers of change with hydrological trends in the TG Halli watershed (the northern portion of the Arkavathy watershed). This region coincides with land use maps recently developed for 1973-74, 1991-92, 2001-02, and 2013-14. Using a linear regression of the hydrological trends on land use, we found the streamflow decline was closely associated with groundwater irrigated agriculture. As we describe in the revised manuscript, the mechanisms of change are likely related to groundwater pumping and construction of in-stream check dams. There was not a clear relationship between streamflow decline and Eucalyptus plantations. Based on referee comments and this new analysis, we have modified the introduction and discussion sections to include a more detailed analysis of the relationship between hydrological change and human drivers of change.

A number of other changes to the manuscript are listed below. We again thank the referees for their consideration of this paper and their helpful comments which we believe has resulted in a better manuscript. We look forward to hearing back regarding the revised manuscript.

Best regards,

Gopal Penny
Veena Srinivasan
Iryna Dronova
Sharad Lele
Sally Thompson

**List of changes to the manuscript**

Abstract
- Minor changes to reflect updates to the manuscript.

Introduction
- Added a paragraph relating the importance of understanding human driven hydrological changes to understanding water resources in South India. This paragraph begins on page 2, line 3.
- Consolidated information related to groundwater irrigation and other potential human drivers of hydrological change, including eucalyptus plantations and check dams. This paragraph begins on page 2, line 30.
- Added a sentence starting on page 3, line 29, describing a new analysis which allows us to associate hydrological trends with land-use (and specifically agricultural practices).

Methods
- Added watershed area to the study site section on page 4, line 3.
- On page 7, line 22, we describe an additional validation approach using Google Earth imagery.
- On page 8, line 35, we describe why we were unable to consider wet season dynamics.
- On page 9, line 18, we provide a more in-depth description of the statistical model coefficient, B.
- On page 9, line 24, we begin a new section entitled Linear regression of hydrological change in land use. This section describes our analysis relating hydrological trends to land-use in the watershed.

Results
- On page 11, line 7, we clarified the meeting of the coefficient B.
- On page 11, line 15, including new section Streamflow decline in agricultural practices. This section describes our ability to predict (in terms of the linear regression) hydrological trends based on agricultural practices.

Discussion
- We rewrote the discussion section Long-term hydrological changes.
- Beginning on page 11, line 25, we describe our use of historical land-use maps and how they allowed us to associate streamflow decline with groundwater irrigated agriculture, while eucalyptus plantations did not have a strong association with streamflow decline.
- On page 12, line 12, we describe the potential influence of check dams on streamflow decline in notes our inability to separate the effects of groundwater pumping and check dams on streamflow decline.
- On page 12, line 26, we describe the downstream wedding affect of urban areas, including changes to water imports and groundwater pumping for the city of Bangalore.

Conclusions
- On page 14, line 1, we begin a new paragraph summarizing our findings related to the heterogeneity of hydrological change, and how we associated hydrological change with groundwater irrigation. We also make suggestions for further investigation.

Additional changes
- We moved the figure of trends and surface water in reservoirs (figure 6 in the original manuscript submission) to supplementary material.
- We removed the figure showing a scatterplot of the magnitude of hydrological change versus agricultural land-use fraction (figure 8 in the original manuscript submission).
- We added a figure (figure 4 in the revised manuscript) showing the divisions of the Arkavathy watershed into sub watersheds and tank cluster watersheds. This figure is referenced on page 8, line 4.
- We added a figure showing trends over time of eucalyptus plantations and irrigated crops in the TG Halli watershed as well as model coefficients from the linear regression relating hydrological change to these two land uses (figure 8 in the revised manuscript). This figure is referenced on page 11, line 19.
- We added a figure to the supplementary material showing a comparison of the water extent in Google Earth images and using our Landsat classification approach. This is figure S6.

**Response to Referee 1**
Referee comments in black
Our (author) response in blue

Strengths

The premise of this paper is interesting and the application of remote sensing to measuring the extent of tank surface area is unique. The paper demonstrates the practical application of remote sensing for characterizing hydrologic change in an otherwise unmonitored setting.

I appreciate the challenge of accounting for various degrees of turbidity in the classification of these water bodies. The methods for measuring tank water extent were clearly presented, and the supplementary figures showing examples of classification were really useful. In my opinion, all of the figures and tables, including those in the supplement, are necessary and contribute to this paper, with the possible exception of Fig. 6. The supplemental tables should make it possible for someone to reproduce this
Analysis.

- We thank the referee for offering careful consideration and analysis of the manuscript. The referee brings up a number of valid points that we believe will strengthen the paper. We intend to incorporate a number of the suggestions from this referee. We have moved Figure 6 to the supplementary material.

Major Concerns

Although I accept that the multiple regression in Eqn. 1 is a reasonable technique to remove precipitation (climate) effects from the estimate of long-term trend, the analysis of hydrologic change related to land use change is not convincing. The visual comparison of percent agriculture with temporal trend in water extent shown in Fig. 8b does not show a clear relationship. It appears that there is only a temporal trend of magnitude greater than 1 ha decadeˆ(-1) 10 kmˆ(-2) (units should be clarified, is this ha/(decade * 10 km^2)?) if the agricultural area is close to 0.75% (which I assume is a typo for 75%); however, low temporal trends are possible for any percent of agricultural land area. This is not a strong argument for a relationship between the two. In fact, the notable negative trends occur only in the two northernmost sub-catchments.

- Thank you for this comment. We have removed that figure (figure 8 in the original submission) from the manuscript. To replace this figure, we've explored a more detailed analysis of the relationship between hydrological trends and land-use. This analysis is described in sections 2.6, 3.3, and 4.1.

An argument could possibly be made that this is an upstream-to-downstream effect, where water withdrawals upstream have a greater impact on stored water over time because return

flows from irrigation dampen the effects of water withdrawals in downstream sub-catchments and/or the major reservoirs shown on Fig. 1 are operated in a way that mitigates long-term trends in water storage changes in the tanks (see for example de Graaf et al., 2014).

- Thank you for this interesting suggestion. On page 12, line 19, we have added a paragraph to the discussion describing our thoughts on upstream-downstream processes.
- Our current hypothesis is that the drying of the northern part of the watershed is linked to groundwater pumping that caused a disconnection between groundwater and surface water (see Srinivasan et al., 2015), leading to reduced baseflow, in a manner analogous to the model in de Graaf et al. (2014). Field studies addressing this mechanism are also currently in progress. Unlike the model presented by de Graaf, however, we are doubtful that the upstream-to-downstream effect is important in the Arkavathy today. Various sources of indirect evidence indicate that the water table is hundreds of meters below the surface in northern parts of the Arkavathy watershed (Srinivasan et al., 2015), suggesting that excess infiltration water is likely to move vertically. Similarly, the relief in the watershed is only about 100 m over a distance of 100 km, again promoting vertical groundwater movement and system-wide return flows connecting upstream to downstream are unlikely. We will make a note to this effect in the revised manuscript.

Additionally, as the authors note on p. 7, lines 28-35, the two watersheds farthest upstream (those that drive the trend) were the only two watersheds with a significant trend in dry season water loss, which they relate to the shift from tank irrigation to groundwater irrigation during the study period. Unless I have misunderstood how dry season losses were treated in the regression, this shift would be reflected in the long-term trend. The authors should test whether or not the change in drying rate is the dominant cause of the trend, and if without this shift, a relationship with the % agricultural area still holds.

- Thank you for this salient observation. We have (briefly) attempted to clarify the following points in the manuscript, beginning on page 9, line 18. Also see page 11, line 7, for a brief discussion around the results.
- Detailed response: non-stationarity in the dry-season water loss term would indeed affect the magnitude of estimated hydrological change in tank clusters given that the regression relationship used to identify this change assumes a stationary loss coefficient.
- The violation of this stationarity assumption in 2 tank clusters might be expected to marginally increase the model error, and, if the time trend in the dry season losses was aligned with the time trend in tank storage, it could indeed confound interpretation of the meaning of the storage trend. However, the trend in dry season losses in the northernmost tank clusters is, instead, in the opposite direction to the trend in storage. Dry-season loss rates have *decreased* over time in the two northernmost subwatersheds. We would expect this change to result in an *increase* in tank water storage after monsoon season (as tanks lose water more slowly). Yet we observe a statistically significant decrease in post-monsoon tank storage over time, in spite of the

decrease in loss rates. Thus, introducing a non-stationary loss coefficient into the model might improve model fit (at the expense of the degrees of freedom of the model) and improve quantitative estimates of the rate of drying due to hydrologic change in the northern watersheds, but would not alter the main conclusion of the study, which is that these watersheds are, in fact, drying.

Are there other spatial patterns in rates of groundwater pumping?

- Understanding the spatial patterns of groundwater extraction in the Arkavathy Basin would be very useful, but unfortunately, monitoring of groundwater use through space has been indirect and sparse. However, groundwater depletion should be associated with groundwater irrigated agriculture (this is not a perfect proxy, as check dams are also likely to be more prevalent in areas with high irrigated agriculture). Please see the discussion, on page 11, line 27, for how we analyze the relationship between hydrological change and irrigated agriculture. In this way, we are essentially using groundwater irrigation agriculture as a proxy for groundwater depletion, with the caveat that we cannot separate groundwater pumping from check dams.

The authors develop a simple mathematical model to extract the trend (B) due to "hydrological change", by which I infer that the authors are referring to the "temporal trends in water extent: : :indicative of long-tem hydrological changes induced by human activity" (p. 3, lines 12-13). The intent would be clearer if the authors were to describe other potential causes of this change (for example, temperature change in the region) and to state when defining B in Eqn. (1) that it is the trend (primarily) due to human-induced hydrological change. Also, because dry season loss is a variable in this regression, it is important that the authors clarify exactly which change B is tracking. As described in lines 27-28, p. 7, the dry season loss term is actually the number of dry season days, rather than a volumetric water loss. As such, the trend B presumably includes year-to-year variations in dry season water use as well. This should be stated explicitly, and instead of loss (L) in Eqn. 1, the authors should refer to the variable as what it is, number of dry season days. In summary, the manuscript needs to be more explicit about what exactly the authors intend B to include and exclude, and why.

- We thank the reviewer for these helpful comments, and particularly the suggestion to frame the response to these issues in terms of the effects on the "meaning" of the trend term B. We have attempted to clarify these issues on page 9, line 18 of the Methods section and beginning page 11, line 7 in the results. We have also changed the designation of dry-season days in the statistical model from "L" to "DSD", which will hopefully help alleviate confusion.
- Detailed response:
- Many of the issues relating to human-induced change (rather than environmental change drivers) were addressed through a hypothesis testing approach in a previous paper (Srinivasan et al. (2015)), which concluded that hydrologic change in the Arkavathy derives from human activity rather than changes in climate or weather.

- We agree that the designation of L as a "loss" term is misleading (as L is the time variable and rather the loss rates arise in the coefficient C,3k). We will consider changing the letter designation of the variable as well as its name in order to clarify the interpretation of that component of the regression.
- We also agree with the reviewer that the magnitude of B could be affected by other sources of variation (that is we are potentially vulnerable to the unobserved variable problem). We note that random interannual variations in water use, dry season losses, evaporative rates etc would not alter the magnitude of B, as they would not change the long-term trend. Rather, random variability would widen the confidence intervals around B.
- Finally, as noted above, we have attempted to clarify in the manuscript the statistical and hydrological interpretation of B. Statistically, B is the temporal trend in total tank water storage over time, after controlling for a stationary relationship between the covariates we describe (Ptotal, Pextreme, L) and tank water storage. Hydrologically, B represents a change in the relationship between both precipitation and dry season water losses and streamflow. Because there is is no change in the effect of dry season water losses in 6/8 watersheds, we interpret B as a change in the rainfall-runoff response. In the two subwatersheds where we detect a change in the effect of dry season water loss on tank storage, we will clarify that B captures the combined effect of hydrological change (streamflow decline pushes B in the negative direction) and dry-season tank water losses (lower tank losses pushes B in a the positive direction). Because B is negative in this area, the effect of hydrological change must exceed that of reduced tank water losses.

Secondary Concerns

The one figure that, to me, is basically a throw away is Fig. 6 for multiple reasons. First, the reservoirs are explicitly exclude from all other parts of the analysis, so whether or not their time-trends are correct is immaterial. Second, the figure does not show an independent source of the temporal evolution of reservoir extent. Third, the conclusion that can be drawn from the satellite imagery matching the timing of reservoir construction is simply that the algorithm can distinguish if, in a very large body of water, there is essentially no water or a lot of it. If this were not the case, there would be no merit in even pursuing this approach at all. It would be reasonable to mention that the method shows the timing of reservoir construction and filling as a single sentence.

- Thank you for this suggestion. We agree with the referee's argument. We have moved Fig 6 to the supplementary material.

In terms of reproducibility, it would be helpful if the authors could provide contact information (an address, perhaps) for Karnataka State Remote Sensing Application Centre as a source for a shapefile of tank boundaries in the Acknowledgments section.

- We have provided the KSNDMC website (see page 11, line 27), which includes their contact information. We are working on publishing a timeseries of Tank water area for all tanks in CUAHSI.

MINOR STUFF:

p. 7, line 20: please clarify why average depth is used for extreme precipitation events rather than total number of extreme events or total depth of precipitation in extreme events.

- We made a minor change on page 8, line 13, noting potential for infiltration excess runoff and the reason for wanting a proxy for rainfall intensity. We did not write a detailed explanation within the manuscripts, hoping that our explanation below will suffice.
- We use average depth of extreme events as a way of approximating heavy rainfall, because our experience in the field suggests a prevalence of infiltration excess runoff. Larger storms are likely to have more infiltration excess runoff due to intense rainfall, and average storm depth is a rough way of approximating this in a way that is feasible (as only daily precipitation data are available) and meets the requirements for the statistical model. Total depth in extreme events is more likely to be correlated with total precipitation depth (and thus add less information to the model) than average storm depth. We will clarify this in the paper.

p. 7, last paragraph: reference Fig. S8.

p. 8, 2nd paragraph: define variable terms explicitly (i.e., The covariates total precipitation, Ptotal,ij, : : :) here, close to the equation, instead of in previous paragraphs. State near the equation that the loss is actually the number of dry season days
- We have defined the variables directly after the equation (page 9, line 11).

p. 8 line 19: clarify what is meant by "centered" (long-term means removed?).
- We will clarify that "centered" entails removing the mean (shifting the data to a mean of zero).

Fig. 7: it would be useful to overlay a drainage/stream map to show how subwatersheds Relate.
- We have added this to Figure 7

p. 10, line 1: clarify what is meant by "The spatial scales of tank clusters are comparable with that of land use"
- This sentence was removed after making changes in the discussion, and hopefully the discussion clarified. What we originally intended to say, is that spatial heterogeneity of hydrological change is important, and that the observed pattern of hydrological change can be related to observed pattern of land use change if we can resolve the hydrological change at a sufficient level of detail.

p. 10, lines 16-17: quotes around "drying" make sense because this is referencing algae blooms giving the false appearance of smaller tank water extent. Quotes around "wetting" do not make sense because the increase in impervious surfaces actually causes tank water extent to increase. It may not be more water in the watershed, but it is more water in the tanks.

- We have removed quotes around wetting in the manuscript (page 12, line 26).

p. 10, line 29: instead of saying ": : :by focusing on land use from a single date.", say ": : :because we only consider land use on [Mon. Day, Year]"

- This sentence was removed from the manuscript.

Figs. S4-S5: at least mention in the caption the water extent vs. precipitation plots.

- We have written a more detailed caption for these figures.

References

de Graaf, I.E.M., L. P. H. van Beek, Y. Wada, M. F. P. Bierkens, 2014: Dynamic attribution of global water demand to surface water and groundwater resources: Effects of abstractions and return flows on river discharges, Adv. Water Resour., 64, 21-33, doi:10.1016/j.advwatres.2013.12.002.

Referee comments in black
Our (author) responses in blue

Overall this is a well written manuscript that attempted to describe trends and spatial differences in changes in hydrology in the Arkavathy watershed on the basis of changes in extracted tank water surface area from satellite images along with other attributes.

- We thank the referee for consideration of our manuscript and valuable advice in helping us clarify some of the key messages of the paper. The referee's feedback has been helpful in alerting us to pieces of writing that need to be improved, particularly in clarifying the broader perspective.

Although the methods were well described, the broader perspective of the analysis is not well presented. After all the study analyzed the tank's surface water dynamics for a very small area (the total area of the Arkavathy is not provided), so, what new information does the findings bring to the community compared to the known facts at regional to national scale for India?

- We have provided the watershed area (4,253 sq. km) in Study Site section. We have updated the introduction to include a new paragraph contextualizing our research in term of water resources in India (see the Introduction, page, 2, line 3). We sought to make the following arguments:
- The Arkavathy contains features that are characteristic of the landscape throughout much of Southern India, and although the findings from our study cannot be directly applied to the region as a whole (given the spatial heterogeneity of the change), the lessons from the Arkavathy can provide clues to hydrologic functioning in the broader region. India faces an array of water scarcity challenges, many of which have been studied at the country scale (Devineni et al., 2013; Tiwari et al., 2009) or at the local field scale (Perrin et al., 2012, Van Meter et al. 2016). Other studies have modeled hydrology at the local scale (Glendenning and Vervoort, 2011) and regional scale (Gosain et al., 2006), but none of these studies describe patterns of surface hydrological change. What is missing from the hydrology literature is an historical analysis at spatial and temporal scales commensurate with the scales of the change. The absence of hydrological records is a primary reason for this gap in the literature (Batchelor et al., 2002; Glendenning et al., 2015), and new datasets are needed that indicate hydrological change at a scale that sufficiently captures the spatial heterogeneity. Such a spatial understanding is particularly pertinent to our study region where the hydrology is truly local, because upstream and downstream subcatchments have been isolated by the fragmentation of the river network (due to tanks and check dams) and the subsurface disconnection due to the vastly depleted groundwater table (as we will clarify in our manuscript, urban effluent can serve to maintain a connected river network directly downstream of urban areas). The heterogeneity of observed changes in the Arkavathy emphasizes one of the problems associated with viewing water trends only at regional or national levels - such large scale trends to not map directly to local scales, yet these are the scales at which people experience and must respond to change. Such local

understanding is of great importance to water managers in southern India, as considerable efforts are underway for river and tank rehabilitation in some areas, without a clear understanding of the mechanisms underlying the historical degradation and loss of water resources (Kumar et al., 2016; Srinivasan et al., 2014).

Given the size of the tanks studied, I would imagine the seasonal water area dynamics will have greater implications than the inter-annual dynamics. The manuscript did not discuss anything on the seasonality for these tanks, or how does that influence the trend?

- We agree that seasonal dynamics are interesting to understand in so far as they indicate the seasonal availability of surface water resources. However, we avoided a detailed description of these dynamics for several reasons. Firstly, since tanks are not widely utilized as a surface water resource throughout the Arkavathy Basin today, the importance of understanding these seasonal dynamics is not so great in the present context as in situations where those surface water stores are relied upon by communities. The importance of the tanks as studied in this paper is as indicators of long-term changes through space in the hydrological dynamics that produce the end of monsoon season storage. Secondly, for pragmatic reasons, it is challenging to study within-year variations other than in the dry season. For approximately 6 months of the year, extensive cloud cover obscures many of the tanks in Landsat images and active radar satellite imagery (which can effectively "see through" clouds) is too coarse to estimate water area in small tanks. We appreciate the referee comments and we will more carefully discuss dry-season dynamics in the manuscript.
- On page 8, line 35 of the revised manuscript, we briefly note that we were unable to consider wet season dynamics.

The manuscript mentioned about differences in water quality, turbidity, vegetation in the water which are influential factors for changes in the reflectance. Even though the DN values were converted to reflectance, the manuscript used only one index (NDWI) to classify water surface area, while there were potentially many other methods or index (Senay et al., 2013) could be used to map water surface correctly, as no one index can cover it all.

- We agree that there is no one method for remote sensing classification of surface water. We selected a simple classification method that was consistent across all Landsat sensors (MSS, TM, ETM, OLI). Our method uses NDWI as an initial classification, and we then apply spectral unmixing using Red, Green, and NIR bands. Although more complex methods have been published, they may not result in a significant improvement in confidence in our model, which we believe is sufficient for our purposes.

While the analysis was performed for the time period between 1972 and 2010 the validation was done for 2014 results. To me validation needs to be done for the time for which the trend analysis is performed (few sample years both wet and dry between 1972 and 2010).

As the study area is so small Google earth might provide good data for validation. Have the authors looked into google earth images as a potential source of validation data?

- We thank the referee for this suggestion. We had conducted an additional validation of our classification approach, which has given us increased confidence in our method. We manually delineated Google Earth images from multiple years (with both "normal" and "wet" precipitation). Our classification compared very well with the Google Earth manual delineation (R-squared of 0.97 in both cases). We were unable to find a suitable Google Earth image to conduct this validation in a dry year. Please see page 27, line 22 in Section 2.4, and page 10, line 26 of Section 3.1 for our approach and the brief results. We have also included a supplementary figure (Figure S6) showing the comparison of tank extent between our Landsat classification and the Google Earth manual delineation.

Page 10 line 5: claims that MK analysis confirms an increase in agricultural land use fraction is related to decrease in tank water storage. How? There is no evidence shown in the manuscript that suggests agricultural land use is increasing. This is vague to me.

- We have restructured this analysis, and will make sure to clarify a number of key points. Agriculture has not expanded so much as it has changed over the course of the study period, and these changes appear to be associated with the drying. We conducted a more sophisticated analysis of the drivers of hydrological trends in the TG Halli watershed (the northern portion of the Arkavathy watershed). This region coincides with land use maps recently developed for 1973-74, 1991-92, 2001-02, and 2013-14. Using a linear regression of the hydrological trends on land use, we found that streamflow decline was closely associated with groundwater irrigated agriculture. As we describe in the revised manuscript, the mechanisms of change are likely related to groundwater pumping and construction of in-stream check dams. There was not a clear relationship between streamflow decline and Eucalyptus plantations. This analysis is described in sections 2.6, 3.3, and 4.1.

Page 10 line 11-12: statement connects with changes in land use and management practice with depleted subsurface stores without providing evidence.

- We have restructured this analysis as well. Please see Figure 8 (top) (referenced in Section 3.3, page 11, line 19, and the dynamics described in more detail in section 4.1) for changes in irrigated agriculture over time and changes in Eucalyptus plantations over time. The time-averaged metric of irrigated agriculture shows a clear relationship with long-term hydrological trends in the TG Halli watershed.

Page 11 line 6-7: Target for classification is to identify water and not water cells, in that case how does incorporation of additional land cover will reduce the classification error?

- Because we are using spectral unmixing, the land class end-member affects the calculated water fraction in each cell. For this reason, having additional (and more precise land classes) could potentially improve classification. We have worked to clarify the discussion in section 4.2.

I think the method used in the manuscript is too simplistic, although producing time- series information of tank water surface area is valuable. I am not sure how much new information has

been brought to the community by this study; therefore I am not convinced that HESS is the right journal for this article.

- We agree that the classification is fairly simple, but overcomes a variety of challenges related to the study, such as the need to incorporate imagery from four Landsat sensors (MSS, TM, ETM, OLI), spectral unmixing in all images, cloud and cloud shadow masking, and the temporal nature of water in tanks (and single image gap filling in SLF-off images). We also note that the classification serves its purpose based on the validation we showed in the manuscript, including the additional Google Earth validation. The overall objectives for the paper (and updated validation information) have been clarified in the updated manuscript, as we describe above and in our previous letter to the editor.

Senay, G.B., Velpuri, N.M., Henok, A., Pervez, M.S., Asante, K.O., Gatarwa, K., Asefa, T., & Jay, A. (2013). Establishing an operational waterhole monitoring system using satellite data and hydrologic modelling: Application in the pastoral regions of East Africa. Pastoralism: Research, Policy and Practice, 3, 20.

[revised manuscript text omitted]

---

## Referee Report (RR1)

Review #2 of Penny, G., V. Srinivasan, I. Dronova, S. Lele, and S. Thompson, Spatial characterization of long-term hydrological change in the Arkavathy watershed adjacent to Bangalore, India, submitted to *Hydrol. Earth Syst. Sci.*, 2017.

This version of the article is much improved, particularly in terms of the description of how the change in relationship between rainfall and streamflow, B, is interpreted and in the additional background providing context on the relationship between Arkavathy and water resources in India more broadly.

I appreciate the authors' attempt to look at the land use change data in a different way; however, I still struggle with the analysis of the relationship between land use and B (change in relationship between rainfall and streamflow). Relating a trend (B) to a time-averaged land use fraction is not intuitive to me. I expect that the authors want to use the trend rather than the time series because the trend, as calculated, excludes interannual variability in precipitation and dry season days. Given the limitations on data availability, I think the approach, with the given caveat of not inferring causation, is acceptable. The number of land use fraction measurements (4) would not support a direct comparison of trends, which I suspect is why the authors use the time-average land use fraction. It is unclear how many points are used in the regression (are all 13 tank clusters used or just 3 (see line 25, p. 9)? are unique values of land use fraction used for each tank cluster?). I'd like to see the plot of average irrigated area vs. B in the supplementary materials. It would be good if the authors could clarify exactly what data went into the regression analysis and also give further justification for this choice of method.

---

## Author Response (AR2)

Dear Dr. Shukla,

Thank you for conveying the second round of referee comments and major revisions to us. We have again made improvements to the manuscript in response to the referee comments. On the next page, we list the most relevant changes to the manuscript and supplementary material, followed by a point-by-point response to the referee comments, followed by a marked-up version of the revised manuscript that highlights (in blue) the changes that we made.

In particular, we have addressed the concerns of the reviewers that precipitation variability could be affecting the estimated trends in tank water extent, and that there was insufficient justification for associating trends in tank inflow to land use change. Following your advice, we believe that much of the reviewer concern arose from lack of clarity around the presentation of the statistical methods in the manuscript. Consequently, we have extensively revised the methods section which now:

(1) Provides an overview of the remote sensing analysis, but has moved many of the specific details to Supplemental Material, considerably shortening this section;
(2) Starting with a tank water balance, presents a detailed rationale for terms that were included or excluded from the linear model, and the justification for modeling the present terms in the way that they have been addressed;
(3) Presents the linear model as the outcome of this water balance argument; and
(4) Provides additional analysis to confirm the importance of accounting for the existence of a trend in tank water extent that derives from drivers other than rainfall. We show that excluding this trend from the model reduces the model performance significantly (F-test $p<3.1\times10^{-11}$) and by a meaningful margin ($r^2=0.68$ when B is included, is 0.58 when B is excluded).

We have also added several new results to the text and figures to the Supplemental Material to address the questions of:
(1) Trends (or, in practice, the lack thereof) in precipitation at basin and smaller scales during the study period (Fig. S11 and S12)
(2) The relative effect sizes of precipitation variability versus a non-precipitation related time trend in tank water extent (Fig. S15).

Several other updates, corrections, and clarifications are provided to address the referees' suggestions. We thank the referees for their consideration of this manuscript, and we look forward to hearing back regarding the revised manuscript.

Best regards,

Gopal Penny
Veena Srinivasan
Iryna Dronova
Sharad Lele
Sally Thompson

**List of relevant changes**

- Moved detailed information about remote sensing from the methods section to Supplemental Material. Similarly, moved detailed information about pixel-scale model validation results to the Supplemental Material.

- Extensively rewrote the methods section to show how water balance reasoning at the tank scale informed the design of the statistical model, and clarifying the focus on and interpretation of the *Year* effect term (B) – and why we interpret this term as a proxy for changes in hydrological processes.

- Using an F-test, we compared the statistical model with a restricted version of the model omitting the *Year* effect term (B). The model including B performed significantly better ($p<3.1\times10^{-11}$) than the restricted model, with better $R^2$ (0.68 with the time trend, 0.58 excluding the time trend) (sections 2.5 and 3.2). This helps confirm the importance of non-precipitation-related changes in explaining the variations in tank areal extents over time.

- Added multiple figures to the supplementary material, demonstrating that we were unable to detect any statistically significant long-term trends in precipitation (at the watershed or tank cluster scales) (Figs. S11 and S12). Showed the variability in tank water extent due to precipitation (holding time constant) and due to the non-rainfall-related temporal trend for each of the tank clusters (Fig. S15).

- Added figure S16, plotting the *Year* effect term B against the time averaged land-use fraction for irrigated crops and eucalyptus plantations.

- Clarified nomenclature referring to different time periods in the year with respect to the monsoon season (section 2.1), including renaming the "post-monsoon" to the "end-of-monsoon" period to be consistent with official definitions of the monsoon season.

- Updated our count of the number of end-of-monsoon (previously "post-monsoon") Landsat images from 18 to 16. The statistical analysis is unchanged (18 years was a miscount of the total number of years input into the original model).

- Fixed inconsistencies in the use of the terms *surface water*, *surface area*, and *tank water extent* – the latter is now used preferably throughout the manuscript.

- In the abstract, noted the how much of the variability in the model is attributed to variability in precipitation.

**Point by point response to referees**

Please note, when we reference text in the manuscript, we refer to the new locations in the underlined manuscript (while leaving the referee comments and references unmodified). Whenever including text from the manuscript, previously-written text is reproduced in blue, and revisions are shown in red.

**Reviewer 1**

*This version of the article is much improved, particularly in terms of the description of how the change in relationship between rainfall and streamflow, B, is interpreted and in the additional background providing context on the relationship between Arkavathy and water resources in India more broadly.*

Thank you.

*I appreciate the authors' attempt to look at the land use change data in a different way; however, I still struggle with the analysis of the relationship between land use and B (change in relationship between rainfall and streamflow). Relating a trend (B) to a time-averaged land use fraction is not intuitive to me. I expect that the authors want to use the trend rather than the time series because the trend, as calculated, excludes interannual variability in precipitation and dry season days. Given the limitations on data availability, I think the approach, with the given caveat of not inferring causation, is acceptable. The number of land use fraction measurements (4) would not support a direct comparison of trends, which I suspect is why the authors use the time-average land use fraction.*

Thank you for this query. There were two motivations for focusing on the time averaged land use fraction. Firstly, as the reviewer notes, the land use maps are not highly resolved in time. Secondly, there is considerable uncertainty in the time needed for a land use change to result in a hydrologic change. Many of the causal connections hypothesized to exist between land use and surface water flow in the Arkavathy Watershed are mediated by groundwater depletion. This is a process associated with non-trivial time lags, meaning that a hydrologic change might significantly lag a land use change, complicating trend analysis. Both of these factors suggest a space-for-time approach, in which the variations in mean land use fraction across all 17 tank clusters for which this information is available are related to the time-trends in tank water level, might be the most straightforward way to initially diagnose the effect of different land uses. We completely agree that a more temporally resolved analysis would be desirable, but is not necessarily straightforward to achieve.

*It is unclear how many points are used in the regression (are all 13 tank clusters used or just 3 (see line 25, p. 9)? Are unique values of land use fraction used for each tank cluster?). I'd like to see the plot of average irrigated area vs. B in the supplementary materials. It would be good if the authors could clarify exactly what data went into the*

*regression analysis and also give further justification for this choice of method.*

We apologize for this lack of clarity. The three subwatersheds upstream of the TG Halli reservoir are used, within which there are a total of 17 tank clusters. We have added a supplementary Figure S16 showing $B_{1,j}$ versus land use fraction for both irrigated crops and Eucalyptus plantations. Text is amended as follows (page 8, line 14-32):

We used four land use maps developed for 1973-74, 1991-92, 2001-02, and 2013-14 (Lele and Sowmyashree, 2016) encompassing the TG Halli watershed, which contains the three subwatersheds upstream of the TG Halli reservoir (TG Halli East, Kumudavathy, and Hesaraghatta) and includes a total of 17 tank clusters. The maps differentiate agricultural land use classes into rainfed crops, irrigated agriculture, and Eucalyptus plantations. Irrigated agriculture in this region is supplied almost exclusively by groundwater, allowing us to test whether groundwater irrigated agriculture, increased water utilization by Eucalyptus plantations (Srinivasan et al., 2015), both, or neither, are associated with the trends in surface flows.

In the early 1970s, rainfed agriculture was the primary land use in the TG Halli watershed. Over the study period, many farmers adopted groundwater irrigation and others converted their fields to *Eucalyptus* plantations, which have the potential to mine shallow groundwater or to significantly reduce deep recharge. These land use changes have the potential to reduce surface water flows by depleting subsurface water availability and baseflow over time, likely resulting in a non-stationary streamflow response. This non-stationarity, in conjunction with the relatively sparse availability of land cover data over time, complicated a direct analysis of land use against tank water level. Instead, a space-for-time approach was used to compare the differences in time-averaged land use across each tank cluster to the differences in tank water level trends inferred for each cluster. We therefore calculate the time-average land use fraction corresponding to irrigated crops $A_{irrigated,avg}$ and *Eucalyptus* plantations $A_{Eucs,avg}$ for each of the 17 tank cluster watersheds and regress $B_{1,j}$ against these land fractions:

$$B_{1,j} = C_{Eucs} A_{Eucs,j} + C_{irrigated} A_{irrigated,j}$$

The coefficients, $C_{Eucs}$ and $C_{irrigated}$, correspond to the sensitivity of hydrological change to time average *Eucalyptus* land cover and irrigated agriculture land cover, across all 17 tank clusters. This analysis is not designed to directly infer causation, but rather to understand associations between streamflow decline and agricultural practices.

**Reviewer 2**
*Authors argued about trends in tank water extent that could not be explained by precipitation (page 7 line 28-29) without providing any evidences.*

Respectfully, this statement is a misinterpretation of what was written, which is not an argument, but a description of the statistical method employed. However, we hope that the intent of the statistical model has been clarified by the reworked methods section. The introduction to the statistical modeling section (page 6 lines 2-4) now states:

The aim of the statistical model is to identify changes in tank water extent that could be attributed to changes in streamflow production in the Arkavathy watershed. To achieve this, the model should control for drivers of water extent variability other than streamflow.

The section then elaborates on how such controls can be put in place by consideration of a tank water balance.

*1. The manuscript does not provide any time series plot of summarized tank water extent extracted from LS by the tank cluster in comparison with rainfall. A quick search reveals a paper by Suresh et al., 2010 that discussed inter-annual variability in rainfall around the Arkavathy watershed. If we assume a similar rainfall regime over Arkavathy, it resembles the tank water extent shown for at least 2 tanks in supplemental figure 4 and 5 for 1994, 2002, 2007 and 2008. Rainfall was well below the mean in 1994 and 2002 corresponding to the lowest water extent while the 2007 and 2008 had good rainfall also reflected in the increased water extent for those tanks compared to 1994 and 2002. It is possible that human activities are playing major role in declining tank water extent, but the temporal patterns also need to be evaluated with respect to changes in precipitation. Here is another example by Subash et al., 2014 where it shows rainfall is declining in Karnataka.*

The original manuscript did not include such a discussion because the linear model controlled for variability in precipitation, and the goal of the study was not to explain all sources of variability in tank water extent, but to isolate a signature of non-stationarity in tank water extent that could be attributed to anthropogenic drivers. While we continue to emphasize the importance of isolating such a signature and understanding its spatial variations, we have now also incorporated precipitation trend analyses.

Results quoted in the main text and figures presented in the Supplemental Material confirm that there are no statistically significant trends in precipitation occurring at whole-of-basin, subwatershed or tank cluster scales (see page 6 line 26 – 31 for discussion of data sources and methods, page 9 lines 22-25 for written results, and Figures S11 and S12 for graphical results).

*2. In validation: adding analysis with respect to google earth images is good, however it provides some confusion, the max water extent shown in supplemental figure 6 does not in line with max extent shown in figure 6 in the main text. So are two sets of tanks presented in these two figures totally different? Speaking of water extent area, unlike what is mentioned in page 10 ln 19 and 22, 25 ha and 2.5 ha are equivalent to 276 and 26 pixels according to 30 m pixel size as suggested in Table 1 and section 2.2. Which one is correct? 25 ha or 27.8 pixels? if latter is true then i think it requires finer resolution than 30 m to effectively identify tanks of that size from the satellite image.*

The tanks validated using Google Earth (Figure S8) are a subset of tanks validated using the LISS image (manuscript Fig. 4). Although it is true that the max water extent of the tanks from the Google Earth validation (2004, 2005, 2009) is greater than the max water extent of tanks from the LISS validation (2014), this is not surprising given the variability of tank water extent.

Thank you for catching the error regarding tank area and pixel count. The corrected manuscript reads (page 9, lines 5-6):

A regression of Landsat extent versus reference extent (Figure 6) for tanks less than 25 hectares (27.8 278 pixels) had a slope of 0.98 and coefficient of determination ($R^2$) of 0.95.

*3. Why the image water extents were not compared against the tank area computed from the shapefile of KSRSAC or tank area from the topographic maps where possible? I would expect, the post-monsoon tank water extent would resemble the maximum water extent possible for each tank as it can be expected to be full after monsoon period.*

There are two primary reasons why we have not conducted such an analysis. (1) We do not have information about the quality of the surveyed boundaries with respect to maximum tank water extent. (2) The tanks do not fill up every year. For any given year, the maximum tank water extent at the end of the monsoon can be substantially different than what might be assessed from mapped tank boundaries.

With these considerations in mind, validating against high resolution imagery allows control on the timing of the observations, the actual water extent in image, and constrains uncertainty in the validation dataset.

*4. The manuscript did not provide proper definition of what they mean by pre-monsoon, monsoon, post-monsoon, wet-monsoon, normal-monsoon (fig S6), dry season, monsoon year (page 8 line 18), normal year, wet year (page 10 line 26-27). In southern India there are two different monsoon seasons; southwest and northeast with two rainfall peaks june-july for southwest monsoon and October for northeast monsoon. The above mentioned terms are inconsistently used throughout the manuscript.*

Thank you for noting these nomenclature issues. We have now added some definitions for clarity, particularly with respect to terms used to define discrete time periods and ensuring we follow the official end of the northeast monsoon in December (instead of November), requiring a change in nomenclature from "post-monsoon" period to "end-of-monsoon" period for the Landsat images used in the study. Even though the official wet season ends in December, little precipitation (<5%) arrives that month.

We have expanded the discussion of climate in the study site description as follows (page 4, lines 17-22):

It has a monsoonal climate and mean annual rainfall of 820 mm. The monsoon season includes the southwest monsoon from June to September and the northeast monsoon from October to December. We therefore refer to several discrete periods of time within the year as the pre-monsoon period, taken as April–May, the wet season or monsoon season, between June and December, the end-of-monsoon period, taken as December–January, and the dry season, January–May. We also refer to the "monsoon year", analogous to the usual concept of the water year, spanning the period from April to March of the following year.

Next, we have added some clarification about use of terms such as normal and wet at the time of use (page 5 lines 27-30):

We also used Digital Globe imagery available from Google Earth (Google Earth, 2016) to assess the validity of the classification in normal (680-955 mm) and wet (>955 mm) precipitation years during the study period. Given the limited availability of these images, we were unable to find a dry-year image (< 680 mm) within the study period that was suitable for comparison with a mostly cloud-free Landsat image.

The thresholds for wet and dry years correspond to the upper (955 mm) and lower quartiles (680 mm) of annual precipitation. We have removed mention of wet and normal monsoons, and now refer to these same periods as normal and wet years.

*5. In the entire manuscript it was never mentioned what was the total water extent derived from images, all the tanks or by tank cluster for each year. Seems like it would be really very small.*

We have included the max water extent for each of the clusters in Table S2 of the supplementary material. The total area is not relevant to the analysis at hand, and thus is not discussed.

*6. Page 4 line 16-17 suggest study focus on Dec and Jan tank water extent estimates, however it is not clear how many data points were used in running the trend analysis. If only Dec and Jan estimates were considered then there would be only 18 data points between 1973 and 2010. Is it enough for a trend analysis? I do not think so.*

The study does focus on a single image for each year. Usable end-of-monsoon images were only available in 16 (rather than 18) years out of the 38 years total in the study period. We update the text on page 5, line 5-9 as follows:

A detailed description of the remote sensing methods employed is provided in the Supplemental Material, Section S1, and the main steps are summarized below. Landsat imagery was used for analyses. Sixteen (16) images taken in December or January between 1973 and 2010 were classified to provide information about end-of-monsoon tank water extent. An additional 32 images were also classified to assist in validation, and to provide information about tank water extent variations during the 10 dry season (see Supplemental Material Fig. S1 and Table S1 for imagery dates).

Naturally, more data would be preferable for developing and testing a model. We have, however, provided confidence intervals (which account for sample size) for each of the model coefficients as well as model performance statistics, which show that the model explains 68% of the variation in the observed tank water extents.

We have also run the regression with and without the presence of the non-rainfall-derived time trend ($B_{1,j}$), and confirmed that the model performance increased meaningfully in terms of the $R^2$ and F statistics when the time trend was included. The differences in the resulting models were statistically significant via the F-test, with the null hypothesis that $B_{1,j}=0$, and alternative hypothesis $B_{1,j} \neq 0$, for at least one value of $j$. The results of the F-test reject the null hypothesis ($p < 3.1$ x $10^{-11}$). The confidence intervals then allow us to determine for which clusters $B_{1,j} \neq 0$.

Thus, we are confident that the trend identified in this study is a meaningful component of the time variation in the tank water extent.

We have now added the following text page 8, lines 1-2:

Model performance was assessed using multiple $R^2$ statistics and significance of the trend slope $B_{1,j}$.

And on page 8, lines 9-12:

Because the value of $B_{1,j}$ is the key result of interest, additional analyses were performed to confirm its importance. Specifically, the model was refit while omitting the *Year* effect $B_{1,j}$ . The performance of the two models (with and without $B_{1,j}$ ) was compared via $R^2$ metrics. The significance of deviations between the two model predictions was tested using an F-test ($H_0 : B_{1,j} = 0$, $H_A : B_{1,j}$ 6= 0, for at least one value of $j$).

At page 10, line 10-13 we have added the following:

We confirmed that the *Year* effect $B_{1,j}$ was important for understanding the variations in tank water extent. Omitting the *Year* effect from the tank water extent model lowered the $R^2$ from 0.68 to 0.58. Furthermore, the model predictions with and without the *Year* effect were significantly different according to the F-test ($p < 3:1$x$10^{-11}$). These results allow us to reject the null hypothesis that $B_{1,j} = 0$, meaning that the *Year* effects could be ignored.

*7. Page 8 line 11-12, surface water extent was strongly related to precipitation metrics, what does this mean?*

This statement simply identifies the period of time over which antecedent precipitation best explained the variations in tank water level (September 1 to the date of the Landsat

image).  The statement is rephrased in the rewritten methods section on page 6, lines 26-31, as:

Variations in P (ii) were accounted for using daily rainfall data from 62 gauges operated by the Karnataka State Natural Disaster Monitoring Centre (KSNDMC). Precipitation trends were analyzed using Mann-Kendall non-parameteric tests. Exploratory analysis at the whole-basin scale indicated that tank water extents were most related to precipitation totals from September 1 to to the date of Landsat image acquisition. Contemporary observations in the Arkavathy watershed suggest that only the largest or most intense storms generate runoff. The average depth of large storms (>10 mm/day) from September 1 to the date of the Landsat image was used as a metric of extreme rainfall occurrence to account for these observations.

*8. Page 8 line 31-34, how could the authors confirm tank water storage which is the volume of water dynamics from surface water extent which is an area.*

Thank you for picking up this loose use of language.  Of course, throughout the paper we interpret the surface area (tank water extent) of the tanks as a proxy for storage, but we should be more explicit.

Firstly, we address this use of the proxy by referring to bathymetric surveys currently in publication at *Water* journal (page 6 lines 4-6):

Bathymetric surveys in the Arkavathy watershed indicate that tank surface area is a function of tank volumetric storage (Young et al., 2017). Thus, a volumetric water balance for a tank can be used to consider the drivers of water extent variability.

Throughout the paper we now take care to refer to "tank water extent" rather than storage.  Finally, we used the results from the bathymetric survey papers to associate scaling of storage volumes in tanks to the scaling of surface area, as follows (page 6 lines 20-23):

Carry-over water extent from 2013 monsoon to the start of the 2014 monsoon was > 25% or approximately >12.5% of post monsoon storage for more than 50% of tank clusters, and >50% or approximately >35% of storage for more than 75% of clusters (storage to volume conversions are based on bathymetric data reported in Young et al., 2017).

*9. Page 11 line 25-26, another example of no evidence. No observed streamflow data or analysis was conducted, but conclusion was drawn for observed streamflow. Throughout the manuscript the satellite derived extracted water extent is used vaguely as proxy to tank water storage and streamflow, which to me is inappropriate and incorrect.*

With respect, the link between the rainfall-controlled trend in tank water extent and non-stationary streamflow is not a vague proxy, but a core inference drawn from the results of the analysis.

However, we appreciate that the way the paper was written asked the reader to draw the threads behind this inference together independently, and that we can clarify the logic by making it explicit. We have focused our re-writing of the methods section around this clarification.

The methods section now starts with a tank water balance, asks where data are available to directly control for variations in the drivers of this water balance, explores how observations can be used to infer stationarity or non-stationarity in other terms, and explains the logic behind converting the water balance and stationarity observations into a suitable linear model. The linear model is then presented. The interpretation of the *Year* effect, $B_{1,j}$ as an indicator of non-stationarity in streamflow production is explained explicitly, and additional analysis to confirm the importance of this effect outlined.

Please see the new text on pages 6-8.

*10. Again out of context discussion of water table and drilling of wells in Page 11 line 26-30.*

With respect, this statement is not out of context.

Rather, it again concerns key inferences made in the study.

The land use change analysis explicitly considers irrigated agriculture and eucalyptus plantation area. The hypothesized link between these land uses and declining trends in tank water area as a proxy for streamflow is described in the Methods section, page 8 line 20 – 23:

In the early 1970s, rainfed agriculture was the primary land use in the TG Halli watershed. Over the study period, many farmers adopted groundwater irrigation and others converted their fields to Eucalyptus plantations, which have the potential to mine shallow groundwater or to significantly reduce deep recharge. These land use changes have the potential to reduce surface water flows by depleting subsurface water availability and baseflow over time.

The information provided about the depth to which groundwater wells are now drilled is consistent with this hypothesized link: deep well depths imply a deep water table, which is consistent with the hypothesis above. It is consistent with the finding that negative non-precipitation-related trends in tank surface area are greatest in those basins where groundwater irrigation practices are more extensive.

*11. Page 12 line 4-7, no supporting evidence provided.*

Thank you, we failed to cross reference the figure that provides this evidence (Figure 6). The results quoted refer to the output of the model described in Section 2.5.

We have clarified this section by making these cross references explicit (page 10 lines 22-25):

The regression of the *Year* effect $B_{1,j}$ on irrigated agriculture and *Eucalyptus* land use areas explained most of the differences in $B_{1,j}$ between tank clusters (R2 = 0.68). The relationship between irrigated crops and $B_{1,j}$ was statistically significant (95% confidence intervals of $C_{irrigated}$ excluded zero), and the relationship with *Eucalyptus* plantations was not statistically significant (Fig. 6).

*12. Page 12 line 19-25, if I understand correctly, what was done is, extract tank surface water extent area from satellite images over time and analyze the change that is present in the surface water extent area, then try to correlate these changes with changes in land use. There were no analyses done with streamflow or tank storage, now the authors started discussing about streamflow and tank storage.*

Please see our responses to point 1 and point 9. Our response to this question is covered by these previous responses.

*13. I find the discussion section 4.1 incoherent given the analyses done. The discussion is all about streamflow while the analyses was on tank surface water extent extraction.*

Please see our responses to point 1 and point 9 which justify why our discussion focuses on streamflow.

*14. Figure 6 last line, that is why more than one index/method is needed to accurately identify tanks from multi-date satellite images.*

This issue is covered extensively in the discussion of validation and quality control for the remote sensing analysis.

*15. The entire conclusion is incoherent and not supported by circumstantial evidence.*

We understand this comment to again refer to the fact that the conclusion discusses an interpretation of the study findings in terms of streamflow. We refer to our responses to questions 1 and 9 to cover this and do not propose any modifications to the conclusions.

*16. What is this supplemental figure 9 for? How do the lines drawn? Why the data points for the available images for December not shown in this plot?*

This figure (now Fig. S10) presents the data referred to in Section 2.4 (page 6 lines 19-25 and page 7 lines 1-7). The figure is cross referenced there in the main text. We were unable to obtain any usable images from December 2013. The lines in (a) are lines of best

fit, showing the trajectory of tank water extent late into the dry season. Furthermore, they represent some of the uncertainty in determining drying rates for the end-of-monsoon period, simply because there are limited images within these months over the course of the study period.

*17. Again what is this supplemental figure 7, 8 and 10 for? No mention of this figure in the main text. Graphs show the data and circumstantial evidence needed to support any conclusion or statement made in the text.*

Supplemental Figure 7 (now Figure S14) is needed to enable Table S3 (which is referred to in the main text) to be interpreted. It offers the key to relate the cluster identifiers to their location in the watershed. We now clarify this in the caption as follows:

Figure S14: Subwatershed names and cluster IDs used in the multiple regression. These identifiers are needed to associate the results in Table S3 with their spatial locations, shown in this figure. The Manchanabele and Harobele subwatersheds here are named for reservoirs within the watershed, which are not located at the subwatershed outlet.

Supplemental Figure 8 (now Fig. S13) provides a standard component of QA/QC for a linear model, namely verification that residuals of the model are distributed normally. We now add brief reference to it in the results section (Page 9, line 26-27):

Model residuals were normally distributed (Figure S13).

Supplemental Figure 10 (now Fig. S9), and shows data that are discussed in the main text in the Results Section 3.1. We now cross reference Figure S9 as follows (page 9 lines 16-20):

Although the time variation in most tanks have not been reported as ground data, trends in water storage over time are widely known for some of the major reservoirs. The TG Halli and Hesaraghatta reservoirs declined from a peak storage in the 1970s to much lower contemporary storage. Large increases in water extent were observed in Manchanabele reservoir, which was constructed in 1993, and Harobele reservoir which was constructed in 2004. These anecdotal trends corroborate our findings for these specific structures (Figure S9).

*18. Very poor figures and maps, especially Figure 1, 4 and 7, no coordinates/index maps.*

We respectfully disagree, but are happy to entertain any changes to these figures as requested by the editor or HESS guidelines.

[revised manuscript text omitted]

---

## Author Response (AR3)

Dear Dr. Shukla,

Thank you for conveying the referee comments and minor revisions to us. We have again made improvements to the manuscript in response to the referee comments. On the next page, we list the most relevant changes to the manuscript and supplementary material, followed by a point-by-point response to the referee comments, followed by a marked-up version of the revised manuscript that highlights (in blue) the changes that we made.

In particular, we have sought to clarify the assumptions that were incorporated into the statistical model of tank water extent. We have also included a new figure that shows the location, start year, and end year of each of the 62 precipitation gauges that were used.

We have also added a "Data sharing" section just before the acknowledgements. We intend to publish the data on hydroshare.org, once the manuscript is approved (at that time we will obtain a DOI and add a citation in the Data sharing section).

Several other updates and clarifications are provided below to address the comments of the referees.  We thank you and the referees for consideration of this manuscript.

Best regards,

Gopal Penny
Veena Srinivasan
Iryna Dronova
Sharad Lele
Sally Thompson

**List of relevant changes**

- Included a new figure in the supplementary material (Fig. S11) showing the locations and start and end years of the 62 rain gauges.
- At the suggestion of Referee 1, we removed a figure from the supplementary material (previously, Fig. S15) which showed the variability in tank water due to precipitation versus the variability due to long-term trends.
- Added a "Data sharing" section, after the Conclusions.

**Point by point response to referees**

Please note, when we reference text in the manuscript, we refer to the new locations in the updated manuscript (while leaving the referee comments and references unmodified). Whenever including text from the manuscript, previously-written text is reproduced in blue, and revisions are shown in red.

**Response to Referee #1**

*This manuscript is easier to follow now that more of the methods relating to image classification have been moved into the supplement. I do not know how HESS feels about supplements that are so long; it is essentially a second paper on data analysis.*

Thank you. We will leave the supplement as is, unless we receive further directions for change.

*p. 2, Line 7: Unclear what is meant by "nonstationary water resources." This can be inferred from the rest of the paper, but at line 7, the reader doesn't know.*

We will change the start of this paragraph to read (p. 2, line 6):

Water scarcity in south India is aggravated by the fact that human activities have shifted or reduced the availability of water resources through inter-basin transfers, artificial conveyance, changes in land use, and irrigation (Mohan and Routray, 2015; Kumar et al., 2005). Effective management of water resources in south India requires better characterization of the changing nature of water resources (Kumar et al., 2005; Milly et al., 2008) and associated human drivers of change (Venot et al., 2007; Falkenmark et al., 2007; Wagener et al., 2010).

*p. 6, lines 11-18: Because points i-v are important for understanding the remainder of lthis section, I recommend simplifying/clarifying this part. Perhaps have one sentence that says "Based on the volumetric water balance, water extent variability derives from precipitation, ..." etc. Then perhaps move p. 7, lines 16-20 directly after that sentence. At which point the authors could explain "In developing this model, we made the following assumptions ..."—and those assumptions come from the point i-v. Or at least instead of saying "The statistical model we developed addressed the above sources of variation by...", clarify that i-v are assumptions. i.e., "We developed a statistical model of tank water extent from these sources of variability based on the following assumptions: ..."*

We rewrote (i)-(v) so that they read as assumptions, and have clarified in the sentence before that they are assumptions. After considering various configurations in terms of the structure of this section (including your suggested edits), we think that it made the most sense to present the information in the order of (a) water balance, (b) assumptions and treatment of water balance terms, (c) further clarification regarding assumptions, (d) introduction of the statistical model. We have updated this paragraph about assumptions to read as follows (p. 6, Line 15):

In order to use a regression model to infer long-term hydrological change using records of water extent and precipitation data, we make the following assumptions to account for each of the terms on the right side of the water balance:

1. the initial storage $S(t_r)$ can be approximated with zero,
2. variations in $P$, and thus their contribution to variations in $Q_{in}$, can be accounted for by including precipitation as a covariate in the model,
3. variations in $Q_{out}$ can be neglected, for two reasons: first, because watershed managers report that tanks rarely overflow, so $Q_{out}$ can reasonably be approximated as ~ 0, and second because any overflow that does occur implies that $S$ is equal to its maximum $S_{max}$, so that variations in overflow cannot contribute to changes in observed S,
4. the sum of *Drainage*, *ET* and *Withdrawal* fluxes can be treated as a stationary cumulative loss term, and
5. any time trends in tank water extent that remain, having accounted for (1)--(4), indicate the presence of non-stationarity in tank water extents that could not be explained by variability in precipitation.

*p. 7, lines 8-15: It seems like this paragraph fits better in the study site section.*

We agree that some of this paragraph could be presented in the Study Site section, and we have incorporated the following text into the Study Site section (p. 4, line 32):
The watershed can be divided into 8 subwatersheds (Fig. 8), which include three major tributaries to the Arkavathy (Kumudavathy, Vrishabhavati, and Suvarnamukhi), and 5 other subwatersheds identified by reservoirs or geographic area (Hesaraghatta, TG Halli East, Manchanabele, Kanakapura, Harobele). The major reservoirs in the watershed differ from the tanks in that they are actively managed, providing water for urban and agricultural water users. For this reason, we focus our analysis of hydrological change on the behavior of tanks.

The remainder of the paragraph relates to how we divided the subwatersheds into clusters, which has more to do with our construction of the model than with the study site. For this reason, we have made only minor changes to the paragraph you referenced (now p. 7, lines 18-25):

All analyses proceeded by considering two spatial scales: 8 subwatersheds and 42 smaller hydrologically-connected subwatershed units, which are referred to as tank "clusters" (Fig. 3). Each cluster contained at least 15 tanks having non-zero water extent in at least 4 end-of-monsoon images. Aggregated tank water extents for each cluster form the basis for statistical analysis. Aggregating data in this way overcomes some of the challenges associated with a relatively short record and frequently dry tanks, while offering enough spatial resolution to identify variability in trends across the Arkavathy watershed. Some tanks were constructed during the study period, and these tanks were excluded from the analysis in any years prior to their construction.

*p. 7, line 20: I like the use of "DSD" for dry season days, and I see that it is defined as a variable, but it would make DSD easier to read in the document if you spell it out, i.e. "denoted DSD for dry season days."*

We have made this modification (p. 7, line 30),

...time delay from the beginning of the end-of-monsoon period (December 1) to the date of Landsat image acquisition, denoted *DSD* for dry season days.

*p. 10, line 6 (and a couple of other spots): B1,j is written as Bi,j. If I am reading equation 2 correctly, the 1 subscript is simply a coefficient identifier. Since there is not a second B coefficient, it would be simpler to drop the first subscript and just have Bj.*

The notation of $B_{1,j}$ is consistent with the numeric identifier on the $C$ coefficients. Our preference would be to keep this notation.

*p. 10, line 17: Cite Figs. S11-S12 for negligible trends in rainfall.*

We have cited these figures, now Figs. S12 & S13 (p. 10, line 30).

*p. 10, line 18: Cite Fig. S10 for trends in dry season tank water loss rates.*

We have cited this figure (p. 10, line 31).

*Replace Fig. 3 with Fig. S14. These figures show essentially the same thing.*

Although these figures are similar, Fig. S14 contains the cluster numbers, which are necessary to interpret Table S3. Because these numbers are superfluous in the manuscript, we prefer the simplicity of the current Fig. 3.

*Fig. S15 could probably be removed and just discussed in the primary manuscript using the figure caption.*

Per your suggestion, we have removed this figure from the supplementary material. There are two main takeaways from the figure. The first (illustrating the total variability due to precipitation versus long-term trends) is already described in the results section (see p. 10, lines 9--14). The other key takeaway from this figure is that the variability of tank water extent due to precipitation is fairly similar across all clusters, while the variability of tanks water extent due to temporal trends depends on the cluster. To clarify this second point, we have added the following sentence to Section 3.2 (p. 10, line 11):

Variability in tank water extent due to precipitation was fairly similar across clusters, while the variability due to temporal trends varied greatly across clusters.

**Response to Referee #2**

*The manuscript has improved substantially, thanks to the authors for their efforts. The methods section is much more clear let along the analysis of precipitation.*

Thank you.

*1.      I think having 62 rainfall stations over around 4200 km2 area is pretty good. But it is not clear in the manuscript whether or not all the stations had data over the entire analysis period. Was it daily data? It is important to document the data availability of these stations because authors have summarized all the stations data into one single number per year for the entire study area (Figure S11).*

The precipitation records contained daily data. We have added a map of the precipitation gauges used in the study (Fig. S11), and referenced the map within the manuscript as follows (p. 7, line 4):

Variations in P (2) were accounted for using daily rainfall data from 62 gauges, from the Directorate of Economics & Statistics, Government of Karnataka (see Fig. S11 for station coverage).

The map in Fig S11 includes a color key that shows the start and end year of each rain gauge.

*2.      What is the spatial distribution of these stations? Could the station locations be shown on Figure 1 or in a separate figure?*

The high number of gauges means plotting them in Fig. 1 could add confusion to that figure. We decided to show their locations only within Fig. S11.

*3.      Figure S12: Is this from cluster specific stations? Could the number of rainfall stations by cluster be shown on the plot?*

We have added the following sentence to Section 3.2 (p. 7, line 31), which should address your question:

The precipitation variables were calculated for each station, interpolated over the entire watershed using the inverse-distance squared approach, and spatially averaged for each cluster.

The (new) map of rain gauges (Fig. S11) shows the distribution of gauges and where they are located spatially in relation to each cluster.

4.      *Table S3 is very informative. However, in the last column the maximum tank water extent is informative. I think another column is needed in this table showing the minimum tank water extent, I suppose they will not all be zero. It is needed because it can tell how much of an actual change occurred over the time.*

Thanks for this comment.  We understand that the reviewer is keen for us to show how much change is occurring over time on a per-tank basis.  However, the suggested approach is unfortunately likely to be misleading.  As noted (page 10, line 9), approximately 63% of the variation in tank water extent over time is attributable to rainfall variations.  Showing the minimum and maximum extents will primarily show this influence of rainfall, rather than elucidating a time trend.  The change in extent of the tanks is exactly what is measured by coefficient $B_{1,j}$.  Thus, the best way to understand the magnitude of tank area change is both on a tank cluster basis (avoiding the loss of statistical power and dry tank effects, and via the fitted coefficient $B_{1,j}$ (avoiding the impression of misleadingly large changes that are attributable to both rainfall and changes in tank inflows).

5.      *Figure 3 and 5 need to be checked with HESS map guidelines.*

We have reviewed the manuscript preparation guidelines at https://www.hydrology-and-earth-system-sciences.net/for_authors/manuscript_preparation.html, and done our best to ensure that these figures are consistent with HESS guidelines. If there are additional changes that need to be made we would be happy to do so.

[revised manuscript text omitted]

---

## Author Response (AR4)

Dear Dr. Shukla,

Thank you for your examination of our manuscript. We agree with all of your suggestions and have updated the manuscript and supplementary material accordingly, including moving the remote sensing classification flowchart and the pixel-scale validation to the main manuscript. We have also cited a repository on hydroshare.org in which we have placed the tank water extent classification results, as well as covariates for the model from Section 2.4. This repository is cited in Data Sharing (Section 6).

Best regards,
Gopal

**List of changes to the manuscript**

- Moved the remote sensing classification flowchart and the pixel-scale validation to the main manuscript.
- In Section 6 (Data Sharing), cited a repository on hydroshare.org in which we have placed the tank water extent classification results, as well as covariates for the model from Section 2.4.

**Response to editor comments**

*Dear Authors,*

*Thank you for submitting your response to the last round of reviewers comments and revising manuscript. I think the manuscript is close to be ready for publication. I would like to suggest a few minor changes though. Please find them below.*

*Thanks,*
*Shrad*

*(1) Please move Fig. S4 and Fig. S7 to the main manuscript. In my opinion the classification algorithm is a novel contribution of this study and it's good to emphasize that and also provide details on the method and validation. I would however request to carefully review the corresponding texts (texts already in the main manuscript and new texts to be moved to the main manuscript) to make sure there is no redundancy, and that the text is as concise as possible.*

We have moved both of these figures to the main manuscript. What was Fig. S4 is now Fig. 4, appearing in the methods (Section 2.2) along with the corresponding text. What was Fig. S7 is now Fig. 5, appearing in the Results (Section 3.1).

*(2) Fig. 5 Caption: Please change "crops based from the" to "crops based on the".*
*(3) Page 4 Line 2: Change Figure 2 to Fig. 2.*
*(4) Page 4 Line 19-21: Suggest rephrasing "We therefore refer to ....January-May" to "We therefore refer to April-May as the pre-monsoon period, June-December as the wet or monsoon season, December–January as the end-of-monsoon period and January–May as the dry season".*
*(5) Page 4: Line 23: Change "which a peak" to "which peaks".*
*(6) Page 5: Line 10: Don't need to use Sixteen and 16.*
*(7) Page 7: Line 7: "to" appears twice.*
*(8) Page 11 Line 27: Please change "is more correlated" "has higher correlation with"*
*(9) Page 12, Line 13: Please remove 'e.g. see'. Also in the same sentence to you mean "at present" by "today"?*

(2)-(9): Thank you for these suggestions and corrections. We have incorporated all of them. For (9), we changed "today" to "at present".

[revised manuscript text omitted]